# Effective Uncertainty Estimation with Evidential Models for Open-World Recognition

## Abstract

Reliable uncertainty estimation is crucial when deploying a classifier in the wild. In this paper, we tackle the challenge of jointly quantifying in-distribution and out-of-distribution (OOD) uncertainties. To this end, we leverage the second-order uncertainty representation provided by evidential models and we introduce *KLoS*, a Kullback–Leibler divergence criterion defined on the class-probability simplex. By keeping the full distributional information, KLoS captures class confusion and lack of evidence in a single score. A crucial property of KLoS is to be a class-wise divergence measure built from in-distribution samples and to not require OOD training data, in contrast to current second-order uncertainty measures. We further design an auxiliary neural network, *KLoSNet*, to learn a refined criterion directly aligned with the evidential training objective. In the realistic context where no OOD data is available during training, our experiments show that KLoSNet outperforms first-order and second-order uncertainty measures to simultaneously detect misclassifications and OOD samples. When training with OOD samples, we also observe that existing measures are brittle to the choice of the OOD dataset, whereas KLoS remains more robust.

## 1 Introduction

Safety is a major concern in visual-recognition applications such as autonomous driving (McAllister et al., 2017) and medical imaging (Heckerman et al., 1992). However, modern neural networks (NNs) struggle to detect their own misclassifications (Hendrycks & Gimpel, 2017). In addition, when exposed to out-of-distribution (OOD) samples, NNs have been shown to provide over-confident predictions instead of abstaining (Hein et al., 2019; Nguyen et al., 2015). Obtaining reliable estimates of the predictive uncertainty is thus necessary to safely deploy models in open-world conditions (Bendale & Boult, 2015).

Notable progress has been made in NN uncertainty estimation with the renewal of Bayesian neural networks (Gal & Ghahramani, 2016; Maddox et al., 2019) and ensembling (Lakshminarayanan et al., 2017; Ovadia et al., 2019b). These techniques produce a probability density over the predictive categorical distribution obtained from sampling. A recent class of models, coined *evidential* (Malinin & Gales, 2018; Sensoy et al., 2018), proposes instead to explicitly learn the concentration parameters of a Dirichlet distribution over first-order probabilities. They have been shown to improve generalisation (Joo et al., 2020), OOD detection (Nandy et al., 2020) and adversarial attack detection (Malinin & Gales, 2019).

Based on the subjective logic framework (Josang, 2016), evidential models enrich uncertainty representation with evidence information and enable to represent different sources of uncertainty. Conflicting evidence, e.g., class confusion, is characterized by the expectation of the second-order Dirichlet distribution while the distribution spread on the simplex expresses the amount of evidence in a prediction (Shi et al., 2020). These sources of uncertainty are also known as *data* and *model* uncertainty in the machine learning literature (Malinin & Gales, 2019).

For open-world recognition, a model should be equipped with an uncertainty measure that accounts both for class confusion and lack of evidence to detect misclassifications and OOD samples. The predictive entropy and the *maximum class probability* (MCP) targeting total uncertainty actually reduce probability distributions on the simplex to their expected value and compute first-order uncertainty measures (Joo et al., 2020; Sensoy et al., 2018). This causes a significant loss of information, as

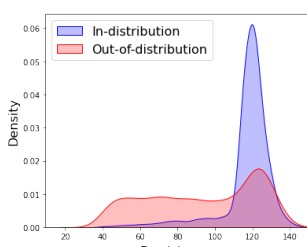

(a) In-distribution image
MCP = 0.50 , entropy = 0.97, KLoS = 97.85

(b) Outlier with same class confusion
MCP = 0.50 , entropy = 0.97, KLoS = 104.71

Figure 1: **Limitations of first-order uncertainty measures and their handling with KLoS**. (a) An in-distribution image with conflicting evidence between *dog* and *wolf*. (b) An outlier with same class confusion but a lower amount of evidence. An evidential neural network (ENN) outputs class-wise evidence information as concentration parameters of a Dirichlet density (visualized on the simplex) over 3-class distributions. Although this density is flatter for the second input, the predictive entropy and MCP, only based on first-order statistics, are equal for both inputs. In contrast, the proposed measure, KLoS, captures both class confusion and lack of evidence, hence correctly reflecting the larger uncertainty for the latter sample.

shown in Fig. 1: the resulting measures are invariant to the spread of the distribution, whereas uncertainty caused by class confusion and lack of evidence should be cumulative, a property naturally fulfilled by the predictive variance in Bayesian regression (Murphy, 2012).

Existing OOD detection methods with evidential models (Malinin & Gales, 2018; 2019; Nandy et al., 2020) use second-order uncertainty measures assuming that the Dirichlet distribution spread is larger for OOD than for in-distribution (ID) samples, e.g., precision $\alpha_0$ or mutual information. They also rely on using auxiliary OOD data during training to enforce higher distribution spread on OOD inputs. However, this assumption is not always fulfilled in absence of OOD training data, as noted by Charpentier et al. (2020); Sensoy et al. (2020). As shown in Fig. 2 for a model trained on CIFAR-10, $\alpha_0$ values largely overlap between IDs and OODs when no OOD training data is used, limiting the effectiveness of existing second-order uncertainty measures. Consequently, neither current first-order nor second-order uncertainty measures appear to be suited for open-world settings.

Figure 2: Precision densities for ID (CIFAR-10) and OOD (TinyImageNet) samples when no OOD training data is used.

**Contributions.** In this paper, we introduce *KLoS*, a measure that accounts for both in-distribution and out-of-distribution sources of uncertainty and that is effective even without having access to auxiliary OOD data at train time. KLoS computes the Kullback–Leibler (KL) divergence between model's predicted Dirichlet distribution and a specifically designed class-wise prototype Dirichlet distribution. By leveraging the second-order uncertainty representation that evidential models provide, KLoS captures both class confusion and lack of evidence in a single score. Prototype distributions are designed with concentration parameters shared with in-distribution training data, which enables to detect OOD samples without assuming any restrictive behavior, e.g., having low precision $\alpha_0$. KLoS naturally reflects the training objective used in evidential models and we propose to learn an auxiliary model, named *KLoSNet*, to regress the values of a refined objective for training samples and to improve uncertainty estimation. To assess the quality of uncertainty estimates in open-world recognition, we design the new task of simultaneous detection of misclassifications and OOD samples. Extensive experiments show the benefits of KLoSNet on various image datasets and model architectures. In presence of OOD training data, we also found that our proposed measure is more robust to the choice of OOD samples while previous measures may perform poorly. Finally, we show that KLoS can be successfully combined with ensembling to improve performance.

## 2 CAPTURING IN-DISTRIBUTION AND OOD UNCERTAINTIES

Section 2.2 presents our measure to capture class confusion and lack of evidence with evidential models. We propose a confidence learning approach to enhance in-distribution uncertainty estimation in Section 2.3. Beforehand, we review evidential models and their learning in Section 2.1 to put in perspective the benefits of the proposed approach.

## 2.1 BACKGROUND: EVIDENTIAL NEURAL NETWORKS

Let us consider a training dataset $\mathcal{D}$ of $N$ *i.i.d.* samples $\boldsymbol{x}$ with label $y \in [\![1, C]\!]$ drawn from an unknown joint distribution $p(\mathbf{x}, \mathbf{y})$. Bayesian models and ensembling methods approximate the posterior predictive distribution $P(\mathbf{y} = c | \boldsymbol{x}^*, \mathcal{D})$ by marginalizing over the network's parameters thanks to sampling. But this comes at the cost of multiple forward passes. Evidential Neural Networks (ENNs) propose instead to model explicitly the posterior distribution over categorical probabilities $p(\boldsymbol{\pi} | \boldsymbol{x}, y)$ by a variational Dirichlet distribution,

$$q_{\boldsymbol{\theta}}(\boldsymbol{\pi} | \boldsymbol{x}) = \mathrm{Dir}\big(\boldsymbol{\pi} | \boldsymbol{\alpha}(\boldsymbol{x}, \boldsymbol{\theta})\big) = \frac{\Gamma(\alpha_0(\boldsymbol{x}, \boldsymbol{\theta}))}{\prod_{c=1}^{C} \Gamma(\alpha_c(\boldsymbol{x}, \boldsymbol{\theta}))} \prod_{c=1}^{C} \pi_c^{\alpha_c(\boldsymbol{x}, \boldsymbol{\theta}) - 1}, \tag{1}$$

whose concentration parameters $\boldsymbol{\alpha}(\boldsymbol{x}, \boldsymbol{\theta}) = \exp f(\boldsymbol{x}, \boldsymbol{\theta})$ are output by a network $f$ with parameters $\boldsymbol{\theta}$; $\Gamma$ is the Gamma function and $\alpha_0(\boldsymbol{x}, \boldsymbol{\theta}) = \sum_{c=1}^{C} \alpha_c(\boldsymbol{x}, \boldsymbol{\theta})$. Precision $\alpha_0$ controls the sharpness of the density with more mass concentrating around the mean as $\alpha_0$ grows. By conjugate property, the predictive distribution for a new point $\boldsymbol{x}^*$ is $P(\mathbf{y} = c | \boldsymbol{x}^*, \mathcal{D}) \approx \mathbb{E}_{q_{\boldsymbol{\theta}}(\boldsymbol{\pi} | \boldsymbol{x}^*)}[\pi_c] = \frac{\exp f_c(\boldsymbol{x}^*, \boldsymbol{\theta})}{\sum_{k=1}^{C} \exp f_k(\boldsymbol{x}^*, \boldsymbol{\theta})}$, which is the usual output of a network $f$ with softmax activation.

The concentration parameters $\boldsymbol{\alpha}$ can be interpreted as pseudo-counts representing the amount of evidence in each class. For instance, in Fig. 1a, the $\boldsymbol{\alpha}$'s output by the ENN indicates that the image is almost equally likely to be classified as *wolf* or as *dog*. More interestingly, it also distinguishes this in-distribution images from the OOD sample in Fig. 1b via the total amount of evidence $\alpha_0$.

**Training Objective.** The ENN training is formulated as a variational approximation to minimize the KL divergence between $q_{\boldsymbol{\theta}}(\boldsymbol{\pi} | \boldsymbol{x})$ and the true posterior distribution $p(\boldsymbol{\pi} | \boldsymbol{x}, y)$:

$$\mathcal{L}_{\mathrm{var}}(\boldsymbol{\theta}; \mathcal{D}) = \mathbb{E}_{(\boldsymbol{x}, y) \sim p(\mathbf{x}, \mathbf{y})} \big[ \mathrm{KL}\big(q_{\boldsymbol{\theta}}(\boldsymbol{\pi} | \boldsymbol{x}) \, \| \, p(\boldsymbol{\pi} | \boldsymbol{x}, y)\big) \big] \tag{2}$$

$$\propto \frac{1}{N} \sum_{(\boldsymbol{x}, y) \in \mathcal{D}} -\big(\psi(\alpha_y(\boldsymbol{x}, \boldsymbol{\theta})) - \psi(\alpha_0(\boldsymbol{x}, \boldsymbol{\theta}))\big) + \mathrm{KL}\big(q_{\boldsymbol{\theta}}(\boldsymbol{\pi} | \boldsymbol{x}) \, \| \, p(\boldsymbol{\pi} | \boldsymbol{x})\big), \tag{3}$$

where $\psi$ is the digamma function. Following Joo et al. (2020), we use the non-informative uniform prior $p(\boldsymbol{\pi} | \boldsymbol{x}) = \mathrm{Dir}\big(\boldsymbol{\pi} | \mathbf{1}\big)$, where $\mathbf{1}$ is the all-one vector, and weigh the KL divergence term with $\lambda > 0$[1]:

$$\mathcal{L}_{\mathrm{var}}(\boldsymbol{\theta}; \mathcal{D}) = \frac{1}{N} \sum_{(\boldsymbol{x}, y) \in \mathcal{D}} -\big(\psi(\alpha_y) - \psi(\alpha_0)\big) + \lambda \mathrm{KL}\big(\mathrm{Dir}(\boldsymbol{\pi} | \boldsymbol{\alpha}) \, \| \, \mathrm{Dir}(\boldsymbol{\pi} | \mathbf{1})\big). \tag{4}$$

In particular, minimizing loss Eq. (4) enforce training sample's precision $\alpha_0$ to remain close to $C + \lambda^{-1}$ value (Malinin & Gales, 2019).

## 2.2 A KULLBACK–LEIBLER DIVERGENCE MEASURE ON THE SIMPLEX

By explicitly learning a distribution of the categorical probabilities $\boldsymbol{\pi}$, evidential models provide a second-order uncertainty representation where the expectation of the Dirichlet distribution relates to class confusion and its spread to the amount of evidence. While originally used to measure the total uncertainty, the predictive entropy $\mathcal{H}[y | \boldsymbol{x}, \boldsymbol{\theta}]$ and the maximum class probability $\mathrm{MCP}(\boldsymbol{x}, \boldsymbol{\theta}) = \max_c P(\mathbf{y} = c | \boldsymbol{x}, \boldsymbol{\theta})$ only account for the position on the simplex. These measures are invariant to the dispersion of the Dirichlet distribution that generates the categorical probabilities. This can be problematic, as illustrated in Fig. 1. To capture uncertainties due to class confusion *and* lack of evidence, an effective measure should account for the sharpness of the Dirichlet distribution and its location on the simplex.

We introduce a novel measure, named *KLoS* for "KL on Simplex", that computes the KL divergence between the model's output and a class-wise prototype Dirichlet distribution with concentrations $\boldsymbol{\gamma}_{\hat{y}}$ focused on the *predicted* class $\hat{y}$:

$$\mathrm{KLoS}(\boldsymbol{x}) \triangleq \mathrm{KL}\Big(\mathrm{Dir}\big(\boldsymbol{\pi} | \boldsymbol{\alpha}(\boldsymbol{x}, \boldsymbol{\theta})\big) \, \| \, \mathrm{Dir}\big(\boldsymbol{\pi} | \boldsymbol{\gamma}_{\hat{y}}\big)\Big), \tag{5}$$

where $\boldsymbol{\alpha}(\boldsymbol{x}, \boldsymbol{\theta}) = \exp f(\boldsymbol{x}, \boldsymbol{\theta})$ are model's output and $\boldsymbol{\gamma}_{\hat{y}} = (1, \dots, 1, \tau, 1, \dots, 1)$ are the uniform concentration parameters except for the predicted class with $\tau = 1 + \lambda^{-1}$.

---

[1] For conciseness, we denote $\alpha_c = \alpha_c(\boldsymbol{x}, \boldsymbol{\theta}), \forall c$ hereafter.

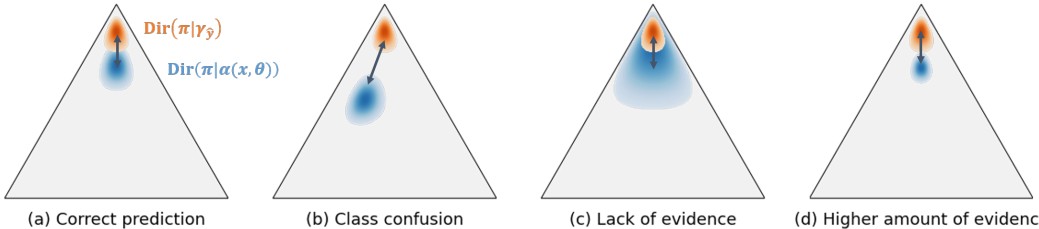

(a) Correct prediction    (b) Class confusion    (c) Lack of evidence    (d) Higher amount of evidence

Figure 3: **KLoS and KLoS$^*$ on the probability simplex**. Given the input sample, the blue region represents the distribution predicted by the evidential model and the orange region represents the prototype Dirichlet distribution with parameters $\gamma_{\hat{y}} = (1, \cdots, 1, \tau, 1, \ldots, 1)$ focused on the predicted class $\hat{y}$. Illustration of the behavior of KLoS in absence of uncertainty (a), in case of class confusion (b) and in case of a different amount of evidence, either lower (c) or higher (d).

The lower KLoS is, the more certain the prediction is. Correct predictions will have Dirichlet distributions similar to the prototype Dirichlet distribution $\gamma_{\hat{y}}$ and will thus be associated with a low uncertainty score (Fig. 3a). Samples with high class confusion will present an expected probability distributions closer to simplex's center than the expected class-wise prototype $p_{\hat{y}}^* = (\frac{1}{K-1+\tau}, \cdots, \frac{\tau}{K-1+\tau}, \cdots, \frac{1}{K-1+\tau})$, resulting in a higher KLoS score (Fig. 3b). Similarly, KLoS also penalizes samples having a different precision $\alpha_0$ than the precision $\alpha_0^* = \tau + C - 1$ of the prototype $\gamma_{\hat{y}}$. Samples with smaller (Fig. 3c) and higher (Fig. 3d) amount of evidence than $\alpha_0^*$ receive a larger KLoS score.

**Effective measure without OOD training data.** Since in-distribution samples are enforced to have precision close to $\alpha_0^*$ during training, the class-wise prototypes are fine estimates of the concentration parameters of training data for each class. Hence, KLoS is a divergence-based metric, which only needs in-distribution data during training to compute its prototypes. This behavior is illustrated in Section 4.1. The proposed measure will be effective to detect various types of OOD samples whose precision is far from $\alpha_0^*$. In contrast, second-order uncertainty measures, e.g., mutual information, assume that OOD samples have smaller $\alpha_0$, a property difficult to fulfill for models trained only with in-distribution samples (see Fig. 2). In Section 4.3, we explore more in-depth the impact of the choice of OOD training data on the actual $\alpha_0$ values for OOD samples.

By approximating the digamma function $\psi$ (see Appendix B), KLoS can also be decomposed as:

$$\text{KLoS}(\boldsymbol{x}) \approx -(\tau - 1) \log\big(\frac{\alpha_{\hat{y}}}{\alpha_0}\big) + \Big( -(\tau - 1)(\frac{1}{2\alpha_0} - \frac{1}{2\alpha_{\hat{y}}}) + \text{KL}\big(\text{Dir}(\boldsymbol{\pi}|\boldsymbol{\alpha}) \,\|\, \text{Dir}(\boldsymbol{\pi}|\mathbf{1})\big)\Big). \quad (6)$$

The first term is the standard log-likelihood and relates only to expected probabilities, hence to the class confusion. The ratio $\alpha_{\hat{y}}/\alpha_0$ makes it invariant to any scaling of the concentration parameters vector $\boldsymbol{\alpha}$. The other terms take into account the spread of the distribution by measuring how close $\alpha_0$ is to $(\tau + C - 1)$, and measure the amount of evidence.

### 2.3 IMPROVING UNCERTAINTY ESTIMATION WITH CONFIDENCE LEARNING

When the model misclassifies an example, i.e., the predicted class $\hat{y}$ differs from the ground truth $y$, KLoS measures the distance between the ENN's output and the wrongly estimated posterior $p(\boldsymbol{\pi}|\boldsymbol{x}, \hat{y})$. This may result in an arbitrarily high confidence / low KL divergence value. Measuring instead the distance to the true posterior distribution $p(\pi|\boldsymbol{x}, y)$ (Fig. 4) would more likely yield a greater value, reflecting the fact that the classifier made an error. Thus, a better measure for misclassification detection would be:

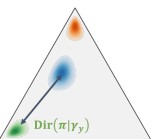

Figure 4: KLoS$^*$

$$\text{KLoS}^*(\boldsymbol{x}, y) \triangleq \text{KL}\Big(\text{Dir}\big(\boldsymbol{\pi}|\boldsymbol{\alpha}(\boldsymbol{x}, \boldsymbol{\theta})\big) \,\|\, \text{Dir}\big(\boldsymbol{\pi}|\gamma_y\big)\Big), \quad (7)$$

where $\gamma_y$ corresponds to the uniform concentrations except for the *true* class $y$ with $\tau = 1 + \lambda^{-1}$.

**Connecting KLoS$^*$ with Evidential Training Objective.** Choosing such value for $\tau$ results in KLoS$^*$ matching the objective function in Eq. (4). This means that KLoS$^*$ is explicitly minimized by the evidential model during training for in-distribution samples. By mimicking the evidential

training objective, we reflect the fact that the model is confident about its prediction if KLoS$^*$ is close to zero. In addition, minimizing the KL divergence between the variational distribution $q_{\boldsymbol{\theta}}(\boldsymbol{\pi}|\boldsymbol{x})$ and the posterior $p(\boldsymbol{\pi}|\boldsymbol{x}, y)$ is equivalent to maximizing the evidence lower bound (ELBO). Hence, a small KLoS$^*$ value corresponds to a high ELBO, which is coherent with the common assumption in variational inference that higher ELBO corresponds to "better" models (Gal, 2016).

Obviously, the true class of an output is not available when estimating confidence on test samples. We propose to learn KLoS$^*$ by introducing an auxiliary confidence neural network, *KLoSNet*, with parameters $\boldsymbol{\omega}$, which outputs a confidence prediction $C(\boldsymbol{x}, \boldsymbol{\omega})$. KLoSNet consists of a small decoder, composed of several dense layers attached to the penultimate layer of the original classification network. During training, we seek $\boldsymbol{\omega}$ such that $C(\boldsymbol{x}, \boldsymbol{\omega})$ is close to KLoS$^*(\boldsymbol{x}, y)$, by minimizing

$$\mathcal{L}_{\text{KLoSNet}}(\boldsymbol{\omega}; \mathcal{D}) = \frac{1}{N} \sum_{(\boldsymbol{x}, y) \in \mathcal{D}} \left\| C(\boldsymbol{x}, \boldsymbol{\omega}) - \text{KLoS}^*(\boldsymbol{x}, y) \right\|^2. \tag{8}$$

KLoSNet can be further improved by endowing it with its own feature extractor. Initialized with the encoder of the classification network, which must remain untouched for not affecting its performance, the encoder of KLoSNet can be fine-tuned along with its regression head. This amounts to minimizing Eq. (8) w.r.t. to both sets of parameters.

The training set for confidence learning is the one used for classification training. In the experiments, we observe a slight performance drop when using a validation set instead. Indeed, when dealing with models with high predictive performance and small validation sets, we end up with fewer misclassification examples than in the train set. At test time, we now directly use KLoSNet's scalar output $C(\boldsymbol{x}, \boldsymbol{\omega}')$ as our uncertainty estimate. As previously, the lower the output value, the more confident the prediction.

## 3 RELATED WORK

**Misclassification Detection.** Several works (Jiang et al., 2018; Corbière et al., 2019; Moon et al., 2020) aim to improve the standard MCP baseline (Hendrycks & Gimpel, 2017) in misclassification detection with NNs. In particular, Corbière et al. (2019) design an auxiliary NN to predict a confidence criterion on training points. We adapt their high-level idea for evidential models to tackle jointly the detection of misclassified in-distribution samples and of OOD samples. With evidential models, Shi et al. (2020) use dissonance in their active learning framework. In contrast to other second-uncertainty measures, dissonance relates to class confusion, but does not acknowledge for the amount of evidence.

**Out-of-Distribution Detection.** Bayesian neural networks (Neal, 1996) and ensembling (Lakshminarayanan et al., 2017) offer a principled approach for uncertainty estimation in which a second-order measure can be derived by measuring the dispersion between individual probabilities vectors. But this comes at the expense of an increased computational cost. Evidential models emulate an ensemble of models using a single network, but usually require OOD samples during training (Malinin & Gales, 2018; 2019; Nandy et al., 2020), which may be unrealistic in many applications. KLoS alleviates this constraint and remains effective without OOD in training. Another range of methods proposes to improve OOD detection on any pre-trained model. ODIN (Liang et al., 2018) mitigates over-confidence by post-processing logits with temperature scaling and by adding inverse adversarial perturbations. Lee et al. (2018) proposes a confidence score based on the class-conditional Mahalanobis distance, with the assumption of tied covariance. Although effective, both approaches need OOD data to tune hyperparameters, which might not generalize to other OOD datasets (Shafaei et al., 2019). Finally, Liu et al. (2020) interpret a pre-trained NN as an energy-based model and compute the energy score to detect OOD samples. Interestingly, this score corresponds to the log precision $\log \alpha_0$, which is similar to the EPKL measure Malinin & Gales (2019) used in ENNs.

## 4 EXPERIMENTS

In this section, we assess our method against existing baseline uncertainty measures on synthetic data and we conduct extensive experiments across various image datasets, architectures and settings.

**Experimental Setup.** Uncertainty measures are derived from an evidential model (Eq. (4)) with $\lambda = 10^{-2}$ as in (Joo et al., 2020; Malinin & Gales, 2019), except for second-order metrics where

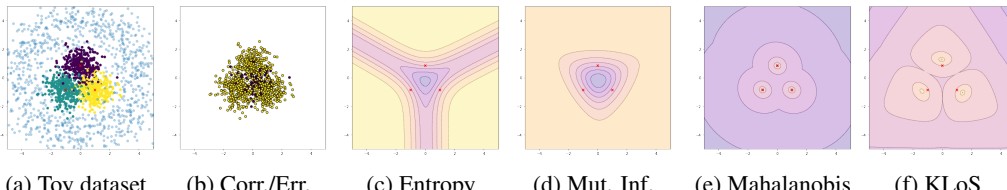

| (a) Toy dataset | (b) Corr./Err. | (c) Entropy | (d) Mut. Inf. | (e) Mahalanobis | (f) KLoS |

Figure 5: **Comparison of various uncertainty measures for a given evidential classifier on a toy dataset.** (a) Training samples from 3 input Gaussian distributions with large overlap (hence class confusion) and OOD test samples (blue); (b) Correct (yellow) and erroneous (red) class predictions on in-domain test samples; (c-f) Visualisation of different uncertainty measures derived from the evidential model trained on the toy dataset. Yellow (resp. purple) indicates high (resp. low) certainty.

we found that setting $\lambda = 10^{-3}$ improves performance. We rely on the learned classifier to train our auxiliary confidence model KLoSNet, using the same training set and following loss Eq. (8). All details about architectures, training algorithms and datasets are available in Appendix C.

**Baselines.** We evaluate our approach against: first-order uncertainty metrics (Maximum Class probability (*MCP*) and predictive entropy (*Entropy*)), second-order metrics (mutual information (*Mut. Inf.*) and *dissonance*), post-training methods for OOD detection (*ODIN* and *Mahalanobis*) and for misclassification detection (*ConfidNet*). Except in Section 4.3, we consider setups where no OOD data is available for training. Consequently, the results reported for ODIN and Mahalanobis are obtained without adversarial perturbations, which is also the best configuration for the considered tasks. We indeed show in Appendix D.6 that these perturbations degrade misclassification detection.

## 4.1 Synthetic Experiment

We analyse the behavior of the KLoS measure and the limitations of existing first- and second-order uncertainty metrics on a 2D synthetic dataset composed of three Gaussian-distributed classes with equidistant means and identical isotropic variance (Fig. 5). This constitutes a scenario with high first-order uncertainty due to class overlap. OOD samples are drawn from a ring around the in-distribution dataset and are only used for evaluation. Fig. 5c shows that Entropy correctly assigns large uncertainty along decision boundaries, which is convenient to detect misclassifications, but yields low uncertainty for points far from the distribution. Mut. Inf. (Fig. 5d) have the opposite behavior than desired by decreasing when moving away from the training data. This is due to the linear nature of the toy dataset where models assign higher concentration parameters far from decision boundaries, hence smaller spread on the simplex, as also noted in (Charpentier et al., 2020). Additionally, Mut. Inf. does not reflect the uncertainty caused by class confusion along decision boundaries. Neither Entropy nor Mut. Inf. is suitable to detect OOD samples in this synthetic experiment. In contrast, KLoS allows discriminating both misclassifications and OOD samples from correct predictions as uncertainty increases far from in-distribution samples for each class (Fig. 5f). KLoS measures a distance between the model's output and a class-wise prototype distribution. Here, we can observe that it acts as a divergence-based measure for each class.

We extend the comparison to include Mahalanobis (Fig. 5e), which is a distance-based measure by assuming Gaussian class conditionals on latent representations, here in the input space. However, Mahalanobis does not discriminate points close to the decision boundaries from points with a similar distance to the origin. Hence, it may be less suited to detect misclassifications than KLoS. Additionally, KLoS does not assume Gaussian distributions in the latent space nor tied covariance, which may be a strong assumption when dealing with high-dimension latent space. In Appendix D.1, a complementary quantitative evaluation on this toy problem confirms our findings regarding the inadequacy of first-order uncertainty measures such as MCP and Entropy, and the improvement provided by KLoS over Mahalanobis on misclassification detection.

## 4.2 Comparative Experiments

The task of detecting both in-distribution misclassifications and OOD samples gives the opportunity to jointly evaluate in-distribution and out-of-distribution uncertainty representations of a method. In this binary classification problem, correct predictions are considered as positive samples while

Table 1: **Comparative experiments on CIFAR-10 and CIFAR-100.** Misclassification (Mis.), out-of-distribution (OOD) and simultaneous (Mis+OOD) detection results (mean % AUROC and std. over 5 runs). Bold type indicates significantly best performance ($p<0.05$) according to paired t-test.

| | Method | Mis. | LSUN | | TinyImageNet | | STL-10 | |
|---|---|---|---|---|---|---|---|---|
| | | | OOD | Mis+OOD | OOD | Mis+OOD | OOD | Mis+OOD |
| **CIFAR-10 VGG-16** | MCP | 87.6 ±1.6 | 79.7 ±1.1 | 84.9 ±1.1 | 80.3 ±1.5 | 85.2 ±1.5 | 60.3 ±1.2 | 75.2 ±1.4 |
| | Entropy | 83.5 ±2.4 | 83.8 ±0.3 | 87.9 ±0.2 | 82.3 ±0.4 | 87.2 ±0.4 | 60.1 ±1.2 | 75.0 ±1.4 |
| | ConfidNet | 90.2 ±0.8 | 82.1 ±1.5 | 87.6 ±1.1 | 83.5 ±0.6 | 88.3 ±0.7 | 61.5 ±1.6 | 77.2 ±1.1 |
| | Dissonance | 91.9 ±0.2 | 84.8 ±0.3 | 90.1 ±0.1 | 84.2 ±0.2 | 89.7 ±0.1 | 64.1 ±0.1 | 79.6 ±0.1 |
| | Mut. Inf. | 84.1 ±1.5 | 84.6 ±0.6 | 85.1 ±1.0 | 80.6 ±0.8 | 83.4 ±1.1 | 61.3 ±0.8 | 65.0 ±2.5 |
| | Diss.+Mut. Inf. | 92.0 ±0.2 | 86.5 ±0.3 | 89.8 ±0.2 | 83.6 ±0.3 | 89.5 ±0.3 | 63.6 ±0.5 | 79.4 ±0.4 |
| | ODIN | 86.0 ±2.0 | 79.5 ±1.2 | 83.8 ±1.5 | 79.6 ±1.9 | 84.0 ±2.0 | 54.7 ±1.5 | 65.0 ±2.6 |
| | Mahalanobis | 91.2 ±0.3 | **88.9** ±0.2 | **91.3** ±0.1 | **86.4** ±0.2 | 90.2 ±0.1 | 63.4 ±0.2 | 78.8 ±0.3 |
| | KLoSNet (Ours) | **92.5** ±0.6 | 87.6 ±0.9 | **91.7** ±0.9 | **86.6** ±0.9 | **91.2** ±0.8 | **67.7** ±1.4 | **81.8** ±0.9 |
| **CIFAR-10 ResNet-18** | MCP | 84.9 ±0.8 | 79.6 ±1.0 | 83.0 ±0.9 | 77.2 ±0.7 | 81.8 ±0.7 | 58.5 ±1.2 | 72.5 ±0.4 |
| | Entropy | 84.6 ±0.8 | 79.6 ±1.1 | 82.8 ±0.9 | 77.2 ±0.7 | 81.6 ±0.7 | 58.4 ±1.2 | 72.2 ±0.4 |
| | ConfidNet | 90.7 ±0.4 | 84.6 ±1.1 | 88.6 ±0.6 | 83.5 ±1.1 | 88.0 ±0.6 | 63.2 ±1.2 | 77.9 ±0.5 |
| | Dissonance | 92.9 ±0.4 | 90.3 ±0.4 | 92.7 ±0.4 | 87.7 ±0.3 | 91.4 ±0.3 | 67.3 ±0.5 | 81.2 ±0.4 |
| | Mut. Inf | 80.6 ±0.6 | 77.0 ±1.2 | 79.4 ±0.9 | 74.3 ±0.8 | 78.0 ±0.7 | 56.4 ±1.0 | 69.1 ±0.2 |
| | Diss.+Mut. Inf. | 92.4 ±0.5 | 86.7 ±1.0 | 90.1 ±0.8 | 84.3 ±0.5 | 88.8 ±0.6 | 65.2 ±0.7 | 80.3 ±0.4 |
| | ODIN | 83.7 ±0.7 | 78.9 ±1.0 | 81.9 ±0.9 | 76.5 ±0.7 | 80.7 ±0.7 | 57.9 ±1.2 | 71.5 ±0.4 |
| | Mahalanobis | 91.2 ±0.4 | 90.7 ±0.4 | 91.8 ±0.3 | 87.6 ±0.4 | 90.3 ±0.4 | 66.8 ±0.5 | 80.0 ±0.3 |
| | KLoSNet (Ours) | **93.9** ±0.4 | **93.1** ±1.1 | **94.4** ±0.3 | **90.6** ±0.6 | **93.2** ±0.2 | **68.5** ±0.3 | **82.3** ±0.2 |
| **CIFAR-100 VGG-16** | MCP | 82.9 ±0.8 | 62.8 ±1.3 | 77.6 ±0.9 | 72.0 ±0.5 | 81.8 ±0.7 | 69.7 ±0.7 | 80.9 ±0.7 |
| | Entropy | 82.2 ±0.8 | 63.2 ±1.4 | 77.2 ±1.0 | 72.5 ±0.6 | 81.5 ±0.8 | 70.1 ±0.8 | 80.6 ±0.7 |
| | ConfidNet | 84.4 ±0.6 | 65.3 ±2.0 | 80.0 ±1.3 | 73.8 ±0.6 | 83.7 ±0.7 | 71.5 ±0.6 | 82.7 ±0.3 |
| | Dissonance | 84.1 ±0.4 | 62.5 ±1.4 | 78.7 ±0.8 | 70.3 ±0.4 | 82.5 ±0.4 | 69.3 ±0.4 | 82.2 ±0.4 |
| | Mut. Inf. | 78.9 ±0.8 | 65.6 ±0.7 | 76.2 ±0.9 | 71.8 ±0.2 | 79.1 ±0.4 | 70.1 ±0.6 | 78.5 ±0.6 |
| | Diss.+Mut. Inf. | 84.2 ±0.6 | 65.1 ±0.3 | 80.1 ±0.4 | 70.1 ±0.3 | 82.5 ±0.5 | 69.5 ±0.3 | 82.3 ±0.5 |
| | ODIN | 82.1 ±0.8 | 62.9 ±1.4 | 77.1 ±1.0 | 71.9 ±0.6 | 81.3 ±0.8 | 69.6 ±0.8 | 80.3 ±0.7 |
| | Mahalanobis | 84.0 ±0.2 | **71.1** ±1.0 | 82.4 ±0.5 | **77.0** ±0.5 | 84.9 ±0.3 | **75.4** ±0.3 | 84.3 ±0.5 |
| | KLoSNet (Ours) | **86.7** ±0.4 | 68.4 ±1.1 | **83.0** ±0.6 | 76.4 ±0.4 | **86.4** ±0.4 | 75.0 ±0.5 | **86.0** ±0.4 |
| **CIFAR-100 ResNet-18** | MCP | 84.0 ±0.4 | 70.4 ±0.9 | 81.0 ±0.3 | 76.6 ±0.5 | 83.6 ±0.4 | 75.4 ±0.5 | 83.1 ±0.2 |
| | Entropy | 83.7 ±0.4 | 70.4 ±0.9 | 80.8 ±0.3 | 76.9 ±0.5 | 83.5 ±0.3 | 75.7 ±0.5 | 83.0 ±0.3 |
| | ConfidNet | **87.1** ±0.2 | 73.0 ±1.4 | **84.5** ±0.6 | 79.1 ±0.3 | 86.8 ±0.3 | **78.5** ±0.8 | **86.6** ±0.5 |
| | Dissonance | 86.7 ±0.4 | 72.3 ±0.4 | 84.0 ±0.2 | 75.0 ±0.4 | 85.3 ±0.4 | 74.7 ±0.3 | 85.2 ±0.2 |
| | Mut. Inf | 82.6 ±0.4 | 70.2 ±1.1 | 80.0 ±0.4 | 76.4 ±0.6 | 82.6 ±0.3 | 75.1 ±0.5 | 82.1 ±0.3 |
| | Diss.+Mut. Inf. | 86.5 ±0.4 | 71.8 ±0.8 | 83.6 ±0.5 | 76.1 ±0.3 | 84.7 ±0.4 | 75.2 ±0.5 | 84.6 ±0.3 |
| | ODIN | 83.7 ±0.4 | 70.3 ±0.9 | 80.8 ±0.7 | 76.6 ±0.5 | 83.5 ±0.3 | 75.4 ±0.5 | 83.0 ±0.3 |
| | Mahalanobis | 85.9 ±0.4 | **75.2** ±0.6 | **84.5** ±0.1 | 78.4 ±0.5 | 85.9 ±0.3 | 77.5 ±0.4 | 85.6 ±0.3 |
| | KLoSNet (Ours) | 86.9 ±0.3 | 73.1 ±0.4 | **84.4** ±0.1 | **80.8** ±0.2 | **87.3** ±0.2 | **79.0** ±0.2 | **86.7** ±0.3 |

misclassified inputs and OOD examples constitute negative samples. Following standard practices (Hendrycks & Gimpel, 2017), we use the area under the ROC curve (AUROC) to evaluate threshold-independent performance. Results for other relevant metrics are available in Appendix D.

The models used in the experiments present high predictive performances. Most often, there are much fewer misclassifications in the test set than considered OOD samples. Hence, joint detection performances might be dominated by the evaluation of the quality of OOD detection. To mitigate this unbalance, we propose to consider the following scheme based on oversampling. Let $\mathcal{A}_M$ be the subset of in-distribution test examples that are misclassified by the observed model and $\mathcal{A}_O$ the set of OOD test samples. We randomly sample $\kappa|\mathcal{A}_O|$ points in $\mathcal{A}_M$, with $\kappa = 1$. Supposing $|\mathcal{A}_O| \geq |\mathcal{A}_M|$, this corresponds to oversampling the set of misclassifications. This over-sampled set is then added to the OOD set to form the negative examples for detection training. The set of correct predictions remains the same. We observed that the variance in AUROC due to this sampling is negligible and we report only the mean hereafter.

Experiments are conducted with VGG-16 (Simonyan & Zisserman, 2015) and ResNet-18 (He et al., 2016) architectures on CIFAR-10 and CIFAR-100 datasets (Krizhevsky, 2009). The OOD datasets used for evaluation are LSUN (Yu et al., 2015), TinyImageNet and STL-10 (Coates et al., 2011). Along with simultaneous detection results, we provide separate results for misclassifications detection and OOD detection respectively in Table 1. On OOD detection, Mahalanobis and KLoSNet outperform other methods, including second-order measures. ODIN also fails to deliver here as

Table 2: **Impact of confidence learning.** Comparison of detection performances between KLoS and KLoSNet for CIFAR-10 and CIFAR-100 experiments with VGG-16 architecture.

|  | Method | Mis. | LSUN |  | TinyImageNet |  | STL-10 |  |
|  |  |  | OOD | Mis+OOD | OOD | Mis+OOD | OOD | Mis+OOD |
|---|---|---|---|---|---|---|---|---|
| **CIFAR-10** | KLoS | 92.1 ±0.3 | 86.5 ±0.3 | 91.2 ±0.2 | 85.4 ±0.3 | 90.4 ±0.2 | 64.1 ±0.3 | 79.6 ±0.3 |
| VGG-16 | KLoSNet | **92.5** ±0.6 | **87.6** ±0.9 | **91.7** ±0.9 | **86.6** ±0.9 | **91.2** ±0.8 | **67.7** ±1.4 | **81.8** ±0.9 |
| **CIFAR-100** | KLoS | 85.4 ±0.2 | 65.1 ±1.1 | 81.3 ±0.6 | 74.5 ±0.4 | 85.4 ±0.4 | 72.7 ±0.3 | 84.8 ±0.4 |
| VGG-16 | KLoSNet | **86.7** ±0.4 | **68.4** ±1.1 | **83.0** ±0.6 | **76.4** ±0.4 | **86.4** ±0.4 | **75.0** ±0.5 | **86.0** ±0.4 |

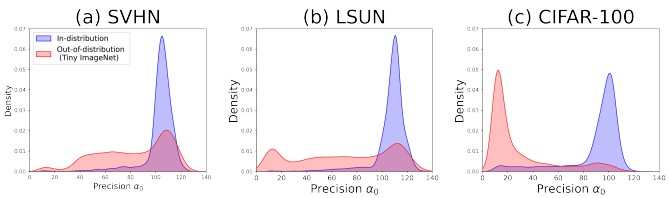

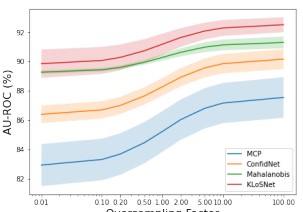

Figure 6: **Effect of OOD training data on precision** $\alpha_0$. Density plots for CIFAR-10/TinyImageNet benchmark: (a,b) with inappropriate OOD samples (SVHN, LSUN); (c) with close OOD samples (CIFAR-100).

Figure 7: **Impact of the oversampling factor** $\kappa$ (CIFAR-10/TinyImageNet).

logits are small due to regularization in the evidential training objective. Mut. Inf. and other spread-based second-order uncertainty measures (see Appendix D.2) fall short to detect correctly OOD. Indeed, for settings where OOD training data is not available, there is no guarantee that every OOD sample will result in lower predicted concentration parameters as previously shown by the density plot of precision $\alpha_0$ in Fig. 2. This stresses the importance of class-wise divergence-based measure. While Mahalanobis may sometimes be slightly better than KLoSNet for OOD detection, it performs significantly less well in misclassification detection, in line with the behavior shown in synthetic experiments. As a result, KLoSNet appears to be the best measure in every simultaneous detection benchmark. For instance, for CIFAR-10/STL-10 with VGG-16, KLoSNet achieves $81.8\%$ AUROC while the second best, Mahalanobis, scores $78.8\%$. We also observe that KLoSNet improves significantly misclassification detection, even compared to dedicated methods such as ConfidNet or second-order measures related to class confusion, e.g., dissonance. Another baseline could be to combine two measures specialized respectively for class confusion and lack of evidence, such as Dissonance+Mut.Inf. But it still performs less well than KLoSNet. In Appendix E, we evaluate our approach on the task of selective classification when inputs are subject to corruptions (which can be seen as OOD samples close to the input distribution, hence particularly challenging), and we observe similar results.

**Impact of Confidence Learning.** To evaluate the effect of the uncertainty measure KLoS and of the auxiliary confidence network KLoSNet, we report a detailed ablation study in Table 2. We can notice that KLoSNet improves misclassification over KLoS but also OOD detection in every benchmark. We intuit that learning to improve misclassification detection also helps to spot some OOD inputs that share similar characteristics.

**Oversampling Factor.** When deploying a model in the wild, it is difficult to know beforehand the proportions of misclassifications and OOD samples it will have to handle. We vary the oversampling factor $\kappa$ in $[0.01; 100]$ for CIFAR10/TinyImageNet in Fig. 7 to assess the robustness of our approach. KLoSNet consistently outperforms all other measures, with a larger gain when $\kappa$ increases.

**Combining KLoS with Ensembling.** Aggregating predictions from an ensemble of neural networks not only improves generalization (Lakshminarayanan et al., 2017; Rame & Cord, 2021) but also uncertainty estimation (Ovadia et al., 2019a). We train an ensemble of ten evidential models on CIFAR-10 and evaluate the performance of KLoS obtained from averaged concentration parameters. On CIFAR-10/TinyImageNet benchmark, misclassification and OOD detection performances are improved by respectively $+1.8$ points and $+1.9$ points, resulting in a $+1.1$ points gain on the joint detection task.

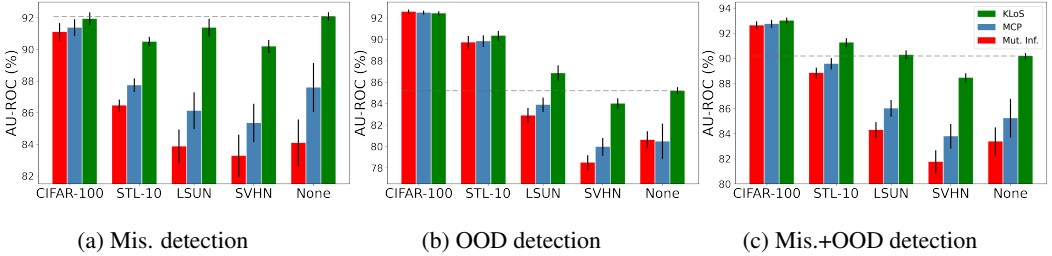

(a) Mis. detection      (b) OOD detection      (c) Mis.+OOD detection

Figure 8: **Comparative detection results with different OOD training datasets.** While using OOD samples in training improves performance in general, the gain value varies widely, sometimes even being negative for inappropriate OOD samples e.g., SVHN. KLoS remains the best measure in every setting. Experiment with VGG-16 architecture on CIFAR-10 dataset.

### 4.3 EFFECT OF TRAINING WITH OUT-OF-DISTRIBUTION SAMPLES

Previous results demonstrate that simultaneous detection of misclassifications and OOD samples can be significantly improved by KLoSNet. We now investigate settings where OOD samples are available. We train an evidential model to minimize the reverse KL-divergence (Malinin & Gales, 2019) between the model output and a sharp Dirichlet distribution focused on the predictive class for in-distribution samples, and between the model output and a uniform Dirichlet distribution for OOD samples. This loss induces low concentration parameters for OOD data and improves second-order uncertainty measures such as Mut. Inf

The literature on evidential models only deals with an OOD training set somewhat related to the in-distribution dataset, e.g. CIFAR-100 for models trained on CIFAR-10. Despite semantic differences, CIFAR-10 and CIFAR-100 images were collected the same way, which might explain the generalisation to other OOD samples in evaluation. Contrarily, CIFAR-10 objects and SVHN street-view numbers differ more for instance. In Fig. 8, we vary the OOD training set and compare the uncertainty metrics taken from the resulting models. As expected, using CIFAR-100 as training OOD data improves performance for every measure (MCP, Mut. Inf. and KLoS). However, the boost provided by training with OOD samples depends highly on the chosen dataset: The performance of Mut. Inf. decreases from 92.6% AUC with CIFAR-100 to 82.9% when switching to LSUN, and even becomes worse with SVHN (78.5%) compared to using no OOD data (80.6%). Indeed, Fig. 6 shows that only the CIFAR-100 dataset seems to be effective to enforce low $\alpha_0$ on unseen OOD samples.

We also note that KLoS outperforms or is on par with MCP and Mut. Inf. in every setting. These results confirm the adequateness of KLoS for simultaneous detection and extend our findings to settings where OOD data is available at train time. Most importantly, using KLoS on models without OOD training data yields better detection performance than other measures taken from models trained with inappropriate OOD samples, here being every OOD dataset other than CIFAR-100.

## 5 DISCUSSION

We propose KLoSNet, an auxiliary model to estimate the uncertainty of a classifier for both in-domain and out-of-domain inputs. Based on evidential models, KLoSNet is trained to predict the KLoS* value of a prediction. By design, KLoSNet encompasses both class confusion and evidence information, which is necessary for open-world recognition. Our experiments extensively demonstrate its effectiveness across various architectures, datasets and configurations, and reveal its class-wise divergence-based behavior. Many real-world applications, e.g., based on active learning or domain adaptation, necessitate to correctly detect both sources of uncertainty. It will be interesting to apply KLoSNet in these contexts. We also show that, far from being the panacea, using training OOD samples depends critically on the choice of these samples for existing uncertainty measures. KLoS, on the other hand, is more robust to this choice and can alleviate their use altogether. While finding suitable OOD samples may be easy for some academic datasets, it may turn more problematic in real-world applications, with the risk of degrading performance with an inappropriate choice. How to build suitable OOD training sets is an important open problem to attack.

## 6 REPRODUCIBILITY STATEMENT

To better understand the link between the evidential training objective and KLoS* criterion stated in Appendix A.2, we provide a detailed derivation in Appendix A. We also elaborate on the theoretical decomposition of KLoS (Eq. (6)) in Appendix B. Regarding experiments reproducibility, Appendix C describes extensively how we generate data in the synthetic experiment (Section 4.1), the image classification datasets used to evaluate our approach including in-distribution and OOD datasets, the implementation and the hyperparameters of the experiments shown in Section 4.2. In particular, we took care in our experiments to perform multiple runs and present mean and std results. Giving access to the code has yet to be authorized internally, but we plan to release the code in the future.

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

## A    LINK BETWEEN KLoS* AND EVIDENTIAL TRAINING OBJECTIVE

In this section, we detail the connection between KLoS* and the evidential training objective. For conciseness, we denote $\alpha_c = \alpha_c(\boldsymbol{x}, \boldsymbol{\theta}), \forall c \in [\![0, C]\!]$ hereafter.

### A.1    DERIVING THE EVIDENTIAL TRAINING OBJECTIVE

We first show the equivalence between Eq. (2) and Eq. (4). Starting from Eq. (2),

$$\mathcal{L}_{\text{var}}(\boldsymbol{\theta}; \mathcal{D}) = \mathbb{E}_{(\boldsymbol{x},y) \sim p(\mathbf{x},\mathbf{y})} \big[ \text{KL}\big( q_{\boldsymbol{\theta}}(\boldsymbol{\pi}|\boldsymbol{x}) \parallel p(\boldsymbol{\pi}|\boldsymbol{x}, y) \big) \big] \tag{9}$$

$$= \frac{1}{N} \sum_{(\boldsymbol{x},y) \in \mathcal{D}} \Big[ \int q_{\boldsymbol{\theta}}(\boldsymbol{\pi}|\boldsymbol{x}) \log \frac{q_{\boldsymbol{\theta}}(\boldsymbol{\pi}|\boldsymbol{x})}{p(\boldsymbol{\pi}|\boldsymbol{x}, y)} \Big] \tag{10}$$

$$= \frac{1}{N} \sum_{(\boldsymbol{x},y) \in \mathcal{D}} \Big[ \int q_{\boldsymbol{\theta}}(\boldsymbol{\pi}|\boldsymbol{x}) \log \frac{q_{\boldsymbol{\theta}}(\boldsymbol{\pi}|\boldsymbol{x})p(y|\boldsymbol{x})}{p(y|\boldsymbol{\pi}, \boldsymbol{x})p(\boldsymbol{\pi}|\boldsymbol{x})} \Big] \tag{11}$$

$$= \frac{1}{N} \sum_{(\boldsymbol{x},y) \in \mathcal{D}} \Big[ \mathbb{E}_{q_{\boldsymbol{\theta}}(\boldsymbol{\pi}|\boldsymbol{x})} \big[ -\log p(y|\boldsymbol{\pi}, \boldsymbol{x}) \big] + \text{KL}\big( q_{\boldsymbol{\theta}}(\boldsymbol{\pi}|\boldsymbol{x}) \parallel p(\boldsymbol{\pi}|\boldsymbol{x}) \big) + \log p(y|\boldsymbol{x}) \Big], \tag{12}$$

where $N = \text{card}(\mathcal{D})$. As the log-likelihood $\log p(y|\boldsymbol{x})$ does not depend on parameters $\boldsymbol{\theta}$,

$$\min_{\theta} \mathcal{L}_{\text{var}}(\boldsymbol{\theta}; \mathcal{D}) = \min_{\theta} \frac{1}{N} \sum_{(\boldsymbol{x},y) \in \mathcal{D}} \Big[ \mathbb{E}_{q_{\boldsymbol{\theta}}(\boldsymbol{\pi}|\boldsymbol{x})} \big[ -\log p(y|\boldsymbol{\pi}, \boldsymbol{x}) \big] + \text{KL}\big( q_{\boldsymbol{\theta}}(\boldsymbol{\pi}|\boldsymbol{x}) \parallel p(\boldsymbol{\pi}|\boldsymbol{x}) \big) \Big]. \tag{13}$$

For a sample $(\boldsymbol{x}, y)$, the reverse cross-entropy term amounts to $\mathbb{E}_{\boldsymbol{\pi} \sim q_{\boldsymbol{\theta}}(\boldsymbol{\pi}|\boldsymbol{x})} \big[ -\log p(y|\boldsymbol{\pi}, \boldsymbol{x}) \big] = -\big( \psi(\alpha_y) - \psi(\alpha_0) \big)$ where $\psi$ is the digamma function. Hence, we recover Eq.(3) of the main paper:

$$\min_{\theta} \mathcal{L}_{\text{var}}(\boldsymbol{\theta}; \mathcal{D}) = \frac{1}{N} \sum_{(\boldsymbol{x},y) \in \mathcal{D}} -\big( \psi(\alpha_y) - \psi(\alpha_0) \big) + \text{KL}(q_{\boldsymbol{\theta}}(\boldsymbol{\pi}|\boldsymbol{x}) \parallel p(\boldsymbol{\pi}|\boldsymbol{x})). \tag{14}$$

Considering that most of the training inputs $\boldsymbol{x}$ are associated with only one observation $y$ in $\mathcal{D}$, we should choose small concentrations parameters $\boldsymbol{\beta}$ for the prior $p(\boldsymbol{\pi}|\boldsymbol{x}) = \text{Dir}(\boldsymbol{\pi}|\boldsymbol{\beta})$ to prevent the resulting posterior distribution $p(\boldsymbol{\pi}|\boldsymbol{x}) = \text{Dir}(\boldsymbol{\pi}|\beta_1, ..., \beta_y + 1, ..., \beta_C)$ from being dominated by the prior. However, this causes exploding gradients in the small-value regimes due to the digamma function, e.g. $\psi'(0.01) > 10^{-4}$.

To stabilize the optimization, we follow Joo et al. (2020) and use the non-informative uniform prior distribution $p(\boldsymbol{\pi}|\boldsymbol{x}) = \text{Dir}(\boldsymbol{\pi}|\mathbf{1})$ where $\mathbf{1}$ is the all-one vector, and we weight the KL-divergence term with $\lambda > 0$:

$$\mathcal{L}_{\text{var}}^{\lambda}(\boldsymbol{\theta}; \mathcal{D}) = \frac{1}{N} \sum_{(\boldsymbol{x},y) \in \mathcal{D}} -\big( \psi(\alpha_y) - \psi(\alpha_0) \big) + \lambda \text{KL}\big( \text{Dir}(\boldsymbol{\pi}|\boldsymbol{\alpha}(\boldsymbol{x}, \boldsymbol{\theta})) \parallel \text{Dir}(\boldsymbol{\pi}|\mathbf{1}) \big). \tag{15}$$

While $\mathcal{L}_{\text{var}}^{\lambda}(\boldsymbol{\theta}; \mathcal{D})$ slightly differs from $\mathcal{L}_{\text{var}}(\boldsymbol{\theta}; \mathcal{D})$, both functions lead to the same optima. Indeed, by considering their gradient, we can show that a local optimum of $\mathcal{L}_{\text{var}}(\boldsymbol{\theta}; \mathcal{D})$ is achieved for a sample $\boldsymbol{x}$ when $\boldsymbol{\alpha}^* = (\beta_1, ..., \beta_y + 1, ..., \beta_C)$ and a local optimum of $\mathcal{L}_{\text{var}}^{\lambda}(\boldsymbol{\theta}; \mathcal{D})$ is $\boldsymbol{\alpha}^{\bullet} = (1, ..., 1 + 1/\lambda, ..., 1)$. Hence, their respective ratio between each element is equal:

$$\forall i, j \in [\![1, C]\!], \frac{\alpha_i^*}{\alpha_j^*} = \frac{\alpha_i^{\bullet}}{\alpha_j^{\bullet}}. \tag{16}$$

### A.2    LINK WITH KLoS*

Let us remind the definition of KLoS* as a KL divergence between the model's output and a class-wise prototype Dirichlet distribution with concentrations $\boldsymbol{\gamma}_{\hat{y}}$ focused on the *true* class $y$:

$$\text{KLoS}^*(\boldsymbol{x}, y) \triangleq \text{KL}\Big( \text{Dir}\big( \boldsymbol{\pi}|\boldsymbol{\alpha}(\boldsymbol{x}, \boldsymbol{\theta}) \big) \parallel \text{Dir}\big( \boldsymbol{\pi}|\boldsymbol{\gamma}_y \big) \Big), \tag{17}$$

where $\boldsymbol{\alpha}(\boldsymbol{x}, \boldsymbol{\theta}) = \exp f(\boldsymbol{x}, \boldsymbol{\theta})$ is the model's output and $\boldsymbol{\gamma}_y = (1, \ldots, 1, \tau, 1, \ldots, 1)$ the vector of uniform concentration parameters except for the true class with $\tau = 1/\lambda + 1$.

The KL divergence between two Dirichlet distributions can be obtained in closed form and KLoS* can be calculated as:

$$\mathrm{KLoS}^*(\boldsymbol{x}, y) = \mathrm{KL}\Big(\mathrm{Dir}\big(\boldsymbol{\pi}|\boldsymbol{\alpha}\big) \parallel \mathrm{Dir}\big(\boldsymbol{\pi}|\boldsymbol{\gamma}_y\big)\Big) \tag{18}$$

$$= \log \Gamma(\alpha_0) - \log \Gamma(C - 1 + 1/\lambda) + \log \Gamma(1 + 1/\lambda) - \sum_{c=1}^{C} \log \Gamma(\alpha_c)$$

$$+ \sum_{c \neq y} \big(\alpha_c - 1\big)\big(\psi(\alpha_c) - \psi(\alpha_0)\big) + \big(\alpha_y - (1 + 1/\lambda)\big)\big(\psi(\alpha_y) - \psi(\alpha_0)\big) \tag{19}$$

On the other hand, the KL-divergence between the model's output and an uniform Dirichlet distribution $\mathrm{Dir}\big(\boldsymbol{\pi}|\mathbf{1}\big)$ reads:

$$\mathrm{KL}\Big(\mathrm{Dir}\big(\boldsymbol{\pi}|\boldsymbol{\alpha}\big) \parallel \mathrm{Dir}\big(\boldsymbol{\pi}|\mathbf{1}\big)\Big) = \log \Gamma(\alpha_0) - \log \Gamma(C) - \sum_{c=1}^{C} \log \Gamma(\alpha_c) + \sum_{c=1}^{C} \big(\alpha_c - 1\big)\big(\psi(\alpha_c) - \psi(\alpha_0)\big). \tag{20}$$

Hence, KLoS* can be written as:

$$\mathrm{KLoS}^*(\boldsymbol{x}, y) = -\frac{1}{\lambda}\big(\psi(\alpha_y) - \psi(\alpha_0)\big) + \mathrm{KL}\big(\mathrm{Dir}(\boldsymbol{\pi}|\boldsymbol{\alpha}) \parallel \mathrm{Dir}(\boldsymbol{\pi}|\mathbf{1})\big)$$

$$+ \big(\log \Gamma(1 + 1/\lambda) - \log \Gamma(C - 1 + 1/\lambda) - \log \Gamma(C)\big). \tag{21}$$

Let us decompose $\mathcal{L}_{\mathrm{var}}^{\lambda}(\boldsymbol{\theta}; \mathcal{D}) = \frac{1}{N} \sum_{(\boldsymbol{x}, y) \in \mathcal{D}} l_{\mathrm{var}}^{\lambda}(\boldsymbol{x}, y, \boldsymbol{\theta})$. We retrieve that $\mathrm{KLoS}^*(\boldsymbol{x}) \propto l_{\mathrm{var}}^{\lambda}(\boldsymbol{x}, y, \boldsymbol{\theta}) + r$ where $r = -\big(\log \Gamma(1 + 1/\lambda) - \log \Gamma(C - 1 + 1/\lambda) - \log \Gamma(C)\big)$ does not depend on the model parameters $\boldsymbol{\theta}$.

In summary, minimizing the evidential training objective $\mathcal{L}_{\mathrm{var}}^{\lambda}(\boldsymbol{\theta}; \mathcal{D})$ is equivalent to minimizing the $\mathrm{KLoS}^*$ value of each training point $\boldsymbol{x}$.

## B DECOMPOSITION OF KLoS

This section provides details of the derivation of Eq. (6). Let us remind the definition of KLoS as a KL divergence between the model's output and a class-wise prototype Dirichlet distribution with concentrations $\boldsymbol{\gamma}_{\hat{y}}$ focused on the *predicted* class $\hat{y}$:

$$\mathrm{KLoS}(\boldsymbol{x}) \triangleq \mathrm{KL}\Big(\mathrm{Dir}\big(\boldsymbol{\pi}|\boldsymbol{\alpha}(\boldsymbol{x}, \boldsymbol{\theta})\big) \parallel \mathrm{Dir}\big(\boldsymbol{\pi}|\boldsymbol{\gamma}_{\hat{y}}\big)\Big), \tag{22}$$

where $\boldsymbol{\alpha}(\boldsymbol{x}, \boldsymbol{\theta}) = \exp f(\boldsymbol{x}, \boldsymbol{\theta})$ is the model's output and $\boldsymbol{\gamma}_{\hat{y}} = (1, ..., \tau, ..., 1)$ the vector of uniform concentration parameters except for the predicted class with $\tau > 1$. For conciseness, we denote $\alpha_c = \alpha_c(\boldsymbol{x}, \boldsymbol{\theta}), \forall c \in [\![0, C]\!]$ hereafter.

Similar to the derivation done in Appendix A.2, KLoS can be derived as:

$$\mathrm{KLoS}(\boldsymbol{x}) = -(\tau - 1)\big(\psi(\alpha_{\hat{y}}) - \psi(\alpha_0)\big) + \mathrm{KL}\big(\mathrm{Dir}(\boldsymbol{\pi}|\boldsymbol{\alpha}) \parallel \mathrm{Dir}(\boldsymbol{\pi}|\mathbf{1})\big), \tag{23}$$

where $\mathbf{1}$ is the all-one vector.

By considering the asymptotic series approximation to the digamma function, $\psi(x) = \log x - \frac{1}{2x} + \mathcal{O}(\frac{1}{x^2})$, the previous expression can be approximated by:

$$\mathrm{KLoS}(\boldsymbol{x}) \approx -(\tau - 1)\log(\frac{\alpha_{\hat{y}}}{\alpha_0}) + \Big(-(\tau - 1)(\frac{1}{2\alpha_0} - \frac{1}{2\alpha_{\hat{y}}})\Big) + \mathrm{KL}\big(\mathrm{Dir}(\boldsymbol{\pi}|\boldsymbol{\alpha}) \parallel \mathrm{Dir}(\boldsymbol{\pi}|\mathbf{1})\big). \tag{24}$$

To gain further intuition about the decomposition, we provide illustrations of the first term (negative log-likelihood, NLL) and the second term in Fig. 9, similar to those shown in Fig. 3 of the paper. We also present quantitative results in Table 3. We observe that the NLL term, which is equivalent to MCP

| Method | Mis. (↑) | OOD (↑) |
|---|---|---|
| NLL term | 80.2 ±1.1 | 15.9 ±0.7 |
| Second term | 48.2 ±4.3 | 98.4 ±0.1 |
| KLoS | 79.4 ±1.2 | 98.8 ±0.3 |

Table 3: Results of KLoS decomposition on synth. data.

measure, helps to detect misclassifications while the second term denotes increasing uncertainty as we move either far away from training data. Hence, by using either the NLL term or the second term, one could distinguish the source of uncertainty if needed.

In particular, we can show that the second term reaches its minimum for $\alpha_0 = \tau + C - 1$ thanks to the following lemma.

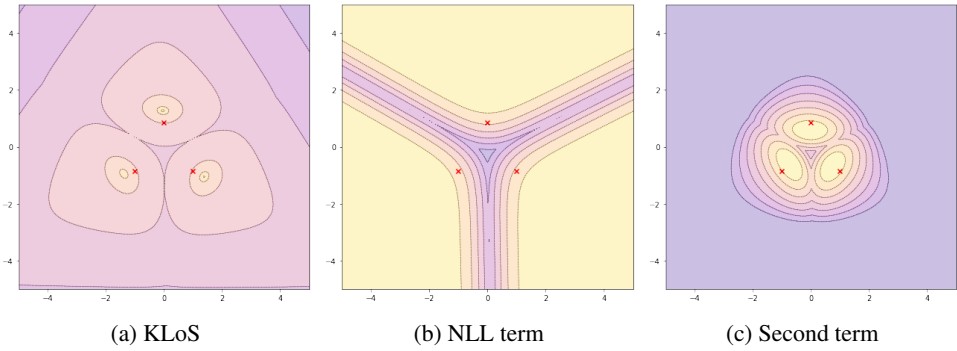

| (a) KLoS | (b) NLL term | (c) Second term |

Figure 9: Visualisation of the different terms of KLoS, derived from the evidential model trained on the same toy dataset than in the main paper.

**Lemma B.1.** *A local minimum of $g : \alpha_0 \to -(\tau - 1)(\frac{1}{2\alpha_0} - \frac{1}{2\alpha_{\hat{y}}})) + \mathrm{KL}\big(\mathrm{Dir}(\boldsymbol{\pi}|\boldsymbol{\alpha}) \parallel \mathrm{Dir}(\boldsymbol{\pi}|\mathbf{1})\big)$ is reached for $\alpha_0 = \tau + C - 1$.*

*Proof.* Let us denote $\boldsymbol{\mu} \in \mathbb{R}^C$ such as $\forall c, \alpha_c = \mu_c \cdot \alpha_0$. By the definition of the KL divergence between two Dirichlet distributions, $g$ reads:

$$g : \alpha_0 \to -(\tau-1)\frac{1-\mu_{\hat{y}}}{2\mu_{\hat{y}}}\frac{1}{\alpha_0} + \log\Gamma(\alpha_0) - \sum_{c=1}^{C}\log\Gamma(\mu_c\alpha_0) + \sum_{c=1}^{C}(\mu_c\alpha_0 - 1)\big(\psi(\mu_c\alpha_0) - \psi(\alpha_0)\big). \tag{25}$$

By taking the derivative:

$$\frac{d}{d\alpha_0}g(\alpha_0) = (\tau-1)\frac{1-\mu_{\hat{y}}}{2\mu_{\hat{y}}}\frac{1}{\alpha_0^2} + \psi(\alpha_0) - \sum_{c=1}^{C}\mu_c\psi(\mu_c\alpha_0) + \sum_{c=1}^{C}\mu_c\big(\psi(\mu_c\alpha_0) - \psi(\alpha_0)\big)$$
$$+ \sum_{c=1}^{C}(\mu_c\alpha_0 - 1)\big(\mu_c\psi^{(1)}(\mu_c\alpha_0) - \psi^{(1)}(\alpha_0)\big), \tag{26}$$

where $\psi^{(1)}(x) = \frac{d}{dx}\psi(x)$ is the trigamma function. As $\sum_{c=1}^{C}\mu_c = 1$, the previous equation simplifies to:

$$\frac{d}{d\alpha_0}g(\alpha_0) = (\tau-1)\frac{1-\mu_{\hat{y}}}{2\mu_{\hat{y}}}\frac{1}{\alpha_0^2} + \sum_{c=1}^{C}(\mu_c\alpha_0 - 1)\big(\mu_c\psi^{(1)}(\mu_c\alpha_0) - \psi^{(1)}(\alpha_0)\big). \tag{27}$$

We consider the asymptotic series approximation to the trigamma function $\psi^{(1)}(x) = \frac{1}{x} + \frac{1}{2x^2} + \mathcal{O}(\frac{1}{x^2})$. Given this approximation, the derivative reads:

$$\frac{d}{d\alpha_0}g(\alpha_0) \approx (\tau-1)\frac{1-\mu_{\hat{y}}}{2\mu_{\hat{y}}}\frac{1}{\alpha_0^2} + \sum_{c=1}^{C}(\mu_c\alpha_0 - 1)\big(\mu_c(\frac{1}{\mu_c\alpha_0} + \frac{1}{2\mu_c^2\alpha_0^2} - (\frac{1}{\alpha_0} - \frac{1}{2\alpha_0^2}) \tag{28}$$

$$= (\tau-1)\frac{1-\mu_{\hat{y}}}{2\mu_{\hat{y}}}\frac{1}{\alpha_0^2} + \sum_{c=1}^{C}(\mu_c\alpha_0 - 1)\frac{1-\mu_c}{2\mu_c}\frac{1}{\alpha_0^2}. \tag{29}$$

The derivative vanishes when $\forall c \neq \hat{y}, \mu_c\alpha_0 = 1$ and $\mu_{\hat{y}}\alpha_0 = \tau$, hence a local minimum of $g$ is reached for $\alpha_0 = \tau + C - 1$. $\qquad\square$

## C EXPERIMENTAL SETUP

In this section, we provide comprehensive details about the datasets, the implementation and the hyperparameters of the experiments shown in Section 4 of the main paper.

**Synthetic Data.** The training dataset (Fig. 3 of the paper) consists of 1,000 samples $(\boldsymbol{x}, y)$ from a distribution $p(\mathbf{x}, \mathbf{y})$ over $\mathbb{R}^2 \times \{1, 2, 3\}$ defined as:

$$p(\mathbf{x} = \boldsymbol{x}, \mathbf{y} = y) = \frac{1}{3}\mathcal{N}(\mathbf{x} = \boldsymbol{x}|\boldsymbol{\mu}_y, \sigma^2 \mathrm{I}_{2\times2}), \tag{30}$$

where $\boldsymbol{\mu}_1 = (0, \sqrt{3}/2)$, $\boldsymbol{\mu}_2 = (-1, -\sqrt{3}/2)$, $\boldsymbol{\mu}_3 = (1, -\sqrt{3}/2)$ and $\sigma = 4$. The marginal distribution of $\mathbf{x}$ is a Gaussian mixture with three equally weighted components having equidistant centers and equal spherical covariance matrices. The test dataset consists of 1,000 other samples from this distribution. Finally, we construct an out-of-distribution (OOD) dataset following Malinin & Gales (2019), by sampling 100 points in $\mathbb{R}^2$ such that they form a 'ring' with large noise around the training points. Some OOD samples will be close to the in-distribution while others will be very far (see Fig. 3 of the paper). The number of OOD samples has been chosen so that it amounts approximately to the number of test points misclassified by the classifier. The classification is performed by a simple logistic regression.

A set of five models is trained for 200 epochs using the evidential training objective (Eq. 6 of the paper) with regularization parameter $\lambda = 5\text{e-}2$ and Adam optimizer with learning rate $0.02$. Uncertainty metrics – MCP, Entropy, Mut. Inf., Malahanobis and KLoS – are computed from these models.

**Image Classification Datasets.** In Sections 4.2 and 4.3 of the main paper, the experiments are conducted using CIFAR-10 and CIFAR-100 datasets Krizhevsky (2009). They consist in $32 \times 32$ natural images featuring 10 object classes for CIFAR-10 and 100 classes for CIFAR-100. Both datasets are composed with 50,000 training samples and 10,000 test samples. We further randomly split the training set to create a validation set of 10,000 images.

OOD datasets are TinyImageNet[2] – a subset of ImageNet (10,000 test images with 200 classes) –, LSUN Yu et al. (2015) – a scene classification dataset (10,000 test images of 10 scenes) –, STL-10 – a dataset similar to CIFAR-10 but with different classes, and SVHN Netzer et al. (2011) – an RGB dataset of $28 \times 28$ house-number images (73,257 training and 26,032 test images with 10 digits) –. We downsample each image of TinyImageNet, LSUN and STL-10 to size $32 \times 32$.

**Training Details.** We implemented in PyTorch Paszke et al. (2019) a VGG-16 architecture Simonyan & Zisserman (2015) in line with the previous works of Charpentier et al. (2020); Malinin & Gales (2019); Nandy et al. (2020), with fully-connected layers reduced to 512 units. Models are trained for 200 epochs with a batch size of 128 images, using a stochastic gradient descent with Nesterov momentum of $0.9$ and weight decay $5\text{e-}4$. The learning rate is initialized at $0.1$ and reduced by a factor of 10 at 50% and 75% of the training progress. Images are randomly horizontally flipped and shifted by $\pm 4$ pixels as a form of data augmentation.

**Balancing Misclassification and OOD Detection.** Most neural networks used in our experiments tend to overfit, which leaves very few training errors available. We provide accuracies on training, validation and test sets in Table 4. With such high predictive performances, the number of misclassifications is usually lower than the number of OOD samples ($\sim$10,000). Hence, the oversampling approach proposed in the paper helps to better balance misclassification detection performances and OOD detection performances in the reported metrics.

|  | CIFAR-10 | CIFAR-100 |
|---|---|---|
| Train | 99.0 $\pm$0.1 | 91.2 $\pm$0.2 |
| Val | 93.6 $\pm$0.1 | 70.6 $\pm$0.3 |
| Test | 93.0 $\pm$0.3 | 70.1 $\pm$0.4 |

Table 4: Mean accuracies (%) and std. over five runs.

**KLoSNet.** We start from the pre-trained evidential model described above. As detailed in Section 3.2 of the main paper, KLoSNet consists of a small decoder attached to the penultimate layer of

---

[2]https://tiny-imagenet.herokuapp.com/

Table 6: **Other baselines results on comparative experiments on CIFAR-10 and CIFAR-100.** Bold type indicates significantly best performance ($p<0.05$) according to paired t-test.

| | | | LSUN | | TinyImageNet | | STL-10 | |
|---|---|---|---|---|---|---|---|---|
| | Method | Mis. | OOD | Mis+OOD | OOD | Mis+OOD | OOD | Mis+OOD |
| **CIFAR-10** VGG-16 | Diff. Ent. | 86.8 ±1.0 | 85.6 ±0.5 | 87.2 ±0.7 | 82.7 ±0.7 | 85.8 ±0.8 | 62.0 ±1.0 | 75.4 ±1.3 |
| | EPKL | 83.9 ±1.5 | 84.5 ±0.7 | 85.1 ±1.0 | 80.4 ±0.8 | 83.2 ±1.2 | 61.3 ±0.8 | 73.8 ±1.1 |
| | KLoSNet (Ours) | **92.5** ±0.6 | 87.6 ±0.9 | **91.7** ±0.9 | **86.6** ±0.9 | **91.2** ±0.8 | **67.7** ±1.4 | **81.8** ±0.9 |
| **CIFAR-10** ResNet-18 | Diff. Ent | 82.7 ±0.6 | 78.3 ±1.2 | 81.1 ±0.9 | 75.9 ±0.8 | 79.9 ±0.7 | 57.5 ±1.1 | 70.8 ±0.3 |
| | EPKL | 80.2 ±0.6 | 76.8 ±1.3 | 79.0 ±0.9 | 74.1 ±0.8 | 77.7 ±0.7 | 56.2 ±1.0 | 68.9 ±0.3 |
| | KLoSNet (Ours) | **93.9** ±0.4 | **93.1** ±1.1 | **94.4** ±0.3 | **90.6** ±0.6 | **93.2** ±0.2 | **68.5** ±0.3 | **82.3** ±0.2 |
| **CIFAR-100** VGG-16 | Diff. Ent. | 80.2 ±0.8 | 65.6 ±0.9 | 77.2 ±0.8 | 72.7 ±0.3 | 80.4 ±0.4 | 71.0 ±0.5 | 79.7 ±0.5 |
| | EPKL | 78.8 ±0.8 | 65.2 ±1.0 | 76.1 ±0.9 | 71.6 ±0.2 | 78.9 ±0.4 | 70.0 ±0.6 | 78.3 ±0.6 |
| | KLoSNet (Ours) | **86.7** ±0.4 | 68.4 ±1.1 | **83.0** ±0.6 | 76.4 ±0.4 | **86.4** ±0.4 | **75.0** ±0.5 | **86.0** ±0.4 |
| **CIFAR-100** ResNet-18 | Diff. Ent | 83.0 ±0.4 | 70.1 ±1.1 | 80.2 ±0.4 | 76.8 ±0.5 | 83.0 ±0.3 | 75.6 ±0.5 | 82.5 ±0.3 |
| | EPKL | 82.5 ±0.4 | 70.2 ±1.1 | 80.0 ±0.4 | 76.3 ±0.6 | 82.5 ±0.3 | 75.0 ±0.5 | 82.0 ±0.2 |
| | KLoSNet (Ours) | **86.9** ±0.3 | 73.1 ±0.4 | **84.4** ±0.1 | **80.8** ±0.2 | **87.3** ±0.2 | **79.0** ±0.2 | **86.7** ±0.3 |

the main network. In CIFAR experiments, this corresponds to VGG-16's fc1 layer of size 512. This auxiliary neural network is composed of five fully-connected layers of size 400, except for the last layer obviously. KLoSNet decoder's weights $\omega$ are trained for 100 epochs with $\ell_2$ loss (Eq. 8 in the main paper) and with Adam optimizer with learning rate 1e-4. As KLoS* ranges from zero to large positive values ($>1000$), one may encounter some issues when training KLoSNet. Consequently, we apply a sigmoid function, $\sigma(x) = \frac{1}{1+e^{-x}}$, after computing the KL-divergence between the NN's output and $\gamma_y$. To prevent over-fitting, training is stopped when validation AUC metric for misclassification detection starts decreasing. Then, a second training step is performed by initializing new encoder $E'$ such that $\theta_{E'} = \theta_E$ initially and by optimizing weights $(\theta_{E'}, \omega)$ for 30 epochs with Adam optimizer with learning rate 1e-6. We stop training once again based on the validation AUC metric.

# D    ADDITIONAL RESULTS

## D.1    DETAILED RESULTS FOR SYNTHETIC EXPERIMENTS

We detail in Table 5 the quantitative results for the task of simultaneous detection of misclassifications and of OOD samples for the synthetic experiment presented in Section 4.1 of the paper. First-order uncertainty measures such as MCP and Entropy perform obviously well on the first task with 80.2% AUC for MCP. However, their OOD performance drops to ~15% AUC on this dataset. On the other hand, Mahalanobis is adapted to detect OOD samples but not as good for misclassifications. KLoS achieves comparable performances to best methods in misclassification detection and in OOD detection (79.4% for Mis. and 98.8% for OOD). As a result, when detecting both inputs simultaneously, KLoS improves all baselines, reaching 89.2% AUC.

| Method | Mis. (↑) | OOD (↑) | Mis+OOD (↑) |
|---|---|---|---|
| MCP | **80.2** ±1.1 | 15.9 ±0.7 | 48.6 ±1.9 |
| Entropy | **78.4** ±1.5 | 11.0 ±0.3 | 45.7 ±1.0 |
| Mut. Inf. | 75.0 ±2.3 | 2.2 ±0.2 | 38.8 ±1.2 |
| Diff. Ent. | 74.2 ±2.7 | 1.9 ±1.0 | 38.0 ±1.3 |
| Mahalanobis | 51.5 ±2.8 | **98.5** ±0.3 | 75.0 ±1.4 |
| KLoS | **79.4** ±1.2 | **98.8** ±0.3 | **89.2** ±0.5 |

Table 5: Synthetic experiment: misclassification (Mis.), out-of-distribution detection (OOD) and simultaneous detection (Mis+OOD) (mean % AUC and std. over 5 runs). Bold type indicates significant top performance ($p < 0.05$) according to paired t-test.

## D.2    RESULTS WITH OTHER SECOND-UNCERTAINTY MEASURES RESULTS

In Table 6, we presents results with other second-uncertainty measures results: differential entropy (*Diff. Ent.*) and *EPKL*. As observed with Mut. Inf. their performances remains below KLoSNet.

Table 7: **AUPR results for experiments with VGG-16 architecture on CIFAR-10 and CI-FAR100**: Misclassification (Mis.), out-of-distribution (OOD) and simultaneous (Mis.+OOD) detection results (% mean and std. over 5 runs).

| | | CIFAR-10 | LSUN | | TinyImageNet | | STL-10 | |
|---|---|---|---|---|---|---|---|---|
| | Method | Mis. (↑) | OOD (↑) | Mis+OOD (↑) | OOD (↑) | Mis+OOD(↑) | OOD (↑) | Mis+OOD(↑) |
| **CIFAR-10**
**VGG-16** | MCP | 44.8 ±2.8 | 80.8 ±0.6 | 92.7 ±0.4 | 80.6 ±0.8 | 92.7 ±0.6 | 57.3 ±0.5 | 85.6 ±0.6 |
| | Entropy | 44.2 ±3.1 | 82.1 ±0.7 | 92.8 ±0.5 | 81.6 ±0.8 | 92.7 ±0.6 | 57.5 ±0.5 | 85.5 ±0.7 |
| | ConfidNet | 45.1 ±3.2 | 81.2 ±1.3 | 93.4 ±0.4 | 82.1 ±0.4 | 93.7 ±0.3 | 58.3 ±0.8 | 86.4 ±0.5 |
| | Mut. Inf | 39.1 ±2.1 | 85.9 ±0.4 | 93.0 ±0.4 | 82.4 ±0.5 | 92.0 ±0.5 | 57.6 ±0.6 | 84.4 ±0.6 |
| | Diff. Ent | 40.8 ±0.3 | 86.5 ±0.3 | 93.7 ±0.3 | 83.6 ±0.4 | 93.0 ±0.3 | 58.4 ±0.4 | 85.3 ±0.5 |
| | EPKL | 39.0 ±2.1 | 85.7 ±0.4 | 92.9 ±0.4 | 82.2 ±0.5 | 91.9 ±0.5 | 57.6 ±0.6 | 84.3 ±0.6 |
| | ODIN | 43.4 ±3.2 | 82.1 ±0.8 | 92.5 ±0.6 | 81.3 ±1.0 | 92.4 ±0.7 | 57.2 ±0.6 | 85.0 ±0.8 |
| | Mahalanobis | 43.7 ±3.5 | **86.6** ±0.4 | **94.9** ±0.3 | **84.9** ±0.3 | 94.4 ±0.2 | 59.5 ±0.2 | 86.9 ±0.4 |
| | KLoSNet (Ours) | **47.6** ±2.5 | 82.2 ±1.8 | **94.8** ±0.2 | 83.1 ±0.7 | **94.7** ±0.2 | **61.7** ±0.6 | **88.4** ±0.6 |
| **CIFAR-100**
**VGG-16** | MCP | 68.2 ±1.6 | 61.4 ±1.1 | 90.6 ±0.6 | 70.3 ±0.7 | 92.5 ±0.4 | 62.5 ±0.6 | 90.3 ±0.4 |
| | Entropy | 67.1 ±1.6 | 62.1 ±1.2 | 90.4 ±0.6 | 71.2 ±0.4 | 92.3 ±0.4 | 63.1 ±0.6 | 90.1 ±0.4 |
| | ConfidNet | 68.5 ±1.8 | 61.7 ±0.6 | 91.0 ±0.4 | 71.2 ±0.4 | 92.9 ±0.4 | 63.7 ±0.5 | 90.8 ±0.3 |
| | Mut. Inf | 63.1 ±1.4 | 63.9 ±0.8 | 89.9 ±0.5 | 70.8 ±0.4 | 91.3 ±0.3 | 63.1 ±0.6 | 89.0 ±0.4 |
| | Diff. Ent | 64.6 ±1.4 | 63.8 ±0.7 | 90.3 ±0.4 | 71.5 ±0.4 | 91.8 ±0.3 | 63.8 ±0.5 | 89.6 ±0.3 |
| | EPKL | 62.9 ±1.4 | 63.9 ±0.8 | 89.8 ±0.5 | 70.7 ±0.4 | 91.2 ±0.3 | 63.0 ±0.6 | 88.9 ±0.4 |
| | ODIN | 67.6 ±1.7 | 61.6 ±1.2 | 90.4 ±0.6 | 70.5 ±0.7 | 92.3 ±0.4 | 62.6 ±0.6 | 90.1 ±0.4 |
| | Mahalanobis | 65.7 ±1.4 | **68.0** ±1.5 | 91.8 ±0.5 | **73.1** ±0.6 | 92.9 ±0.3 | **66.0** ±0.6 | 91.0 ±0.4 |
| | KLoSNet (Ours) | **69.9.** ±1.4 | 64.3 ±1.3 | **92.2** ±0.5 | 72.6 ±0.5 | **93.7** ±0.3 | 65.6 ±0.7 | **92.0** ±0.3 |

(a) CIFAR-10 with VGG-16

| | CIFAR-10 | SVHN | |
|---|---|---|---|
| Method | Mis. (↑) | OOD (↑) | Mis+OOD (↑) |
| MCP | 87.6 ±1.6 | 87.3 ±2.2 | 88.9 ±0.5 |
| Entropy | 83.5 ±2.4 | 85.5 ±2.3 | 86.9 ±1.9 |
| ConfidNet | 90.2 ±0.8 | **89.0** ±3.1 | 91.0 ±1.1 |
| Mut. Inf. | 84.1 ±1.5 | 80.0 ±3.9 | 83.2 ±1.7 |
| Diff. Ent. | 86.8 ±1.0 | 86.0 ±2.0 | 87.6 ±0.9 |
| EPKL | 83.9 ±1.5 | 79.4 ±4.2 | 82.8 ±1.9 |
| ODIN | 86.0 ±2.0 | 86.8 ±2.2 | 87.7 ±1.0 |
| Mahalanobis | 91.2 ±0.3 | **89.1** ±2.8 | 91.5 ±1.1 |
| KLoSNet (Ours) | **92.5** ±0.6 | **89.8** ±3.0 | **92.7** ±1.2 |

(b) CIFAR-10 with ResNet18

| | CIFAR-100 | SVHN | |
|---|---|---|---|
| Method | Mis. (↑) | OOD (↑) | Mis+OOD (↑) |
| MCP | 84.9 ±0.8 | 79.6 ±1.0 | 83.0 ±0.9 |
| Entropy | 84.6 ±0.8 | 79.6 ±1.1 | 82.8 ±0.9 |
| ConfidNet | 90.7 ±0.4 | 84.6 ±1.1 | 88.6 ±0.6 |
| Mut. Inf | 80.6 ±0.6 | 77.0 ±1.2 | 79.4 ±0.9 |
| Diff. Ent | 82.7 ±0.6 | 78.3 ±1.2 | 81.1 ±0.9 |
| EPKL | 80.2 ±0.6 | 76.8 ±1.3 | 79.0 ±0.9 |
| ODIN | 83.7 ±0.7 | 78.9 ±1.0 | 81.9 ±0.9 |
| Mahalanobis | 91.2 ±0.4 | 90.7 ±0.4 | 91.8 ±0.3 |
| KLoSNet (Ours) | **93.9** ±0.4 | **93.1** ±1.1 | **94.4** ±0.3 |

(c) CIFAR-100 with VGG-16

| | CIFAR-10 | SVHN | |
|---|---|---|---|
| Method | Mis. (↑) | OOD (↑) | Mis+OOD (↑) |
| MCP | 82.9 ±0.8 | 70.8 ±3.9 | **81.3** ±2.0 |
| Entropy | 82.2 ±0.8 | **72.9** ±3.9 | **81.5** ±2.0 |
| ConfidNet | 84.4 ±0.6 | 68.0 ±3.4 | 80.8 ±2.0 |
| Mut. Inf. | 78.9 ±0.8 | **72.7** ±4.9 | 79.5 ±2.5 |
| Diff. Ent. | 80.2 ±0.8 | **72.4** ±4.9 | 80.2 ±2.5 |
| EPKL | 78.8 ±0.8 | **72.7** ±4.8 | 79.4 ±2.4 |
| ODIN | 82.1 ±0.8 | 72.0 ±3.8 | **81.3** ±1.9 |
| Mahalanobis | 84.0 ±0.2 | **73.4** ±5.6 | **83.2** ±2.5 |
| KLoSNet (Ours) | **86.7** ±0.4 | 70.4 ±5.7 | **83.5** ±2.8 |

(d) CIFAR-100 with ResNet18

| | CIFAR-100 | SVHN | |
|---|---|---|---|
| Method | Mis. (↑) | OOD (↑) | Mis+OOD (↑) |
| MCP | 84.9 ±0.8 | 79.6 ±1.0 | 83.0 ±0.9 |
| Entropy | 84.6 ±0.8 | 79.6 ±1.1 | 82.8 ±0.9 |
| ConfidNet | 90.7 ±0.4 | 84.6 ±1.1 | 88.6 ±0.6 |
| Mut. Inf | 80.6 ±0.6 | 77.0 ±1.2 | 79.4 ±0.9 |
| Diff. Ent | 82.7 ±0.6 | 78.3 ±1.2 | 81.1 ±0.9 |
| EPKL | 80.2 ±0.6 | 76.8 ±1.3 | 79.0 ±0.9 |
| ODIN | 83.7 ±0.7 | 78.9 ±1.0 | 81.9 ±0.9 |
| Mahalanobis | 91.2 ±0.4 | 90.7 ±0.4 | 91.8 ±0.3 |
| KLoSNet (Ours) | **93.9** ±0.4 | **93.1** ±1.1 | **94.4** ±0.3 |

Figure 10: **Results with SVHN as OOD dataset** (% mean AUROC and std. over 5 runs).

### D.3 AUPR RESULTS WITH VGG-16 ARCHITECTURE

Along with the AUROC metric, we also present results with the area under the precision-recall curve (AUPR) metric in Table 7. The precision-recall (PR) curve is the graph of the precision = $\mathrm{TP}/(\mathrm{TP}+\mathrm{FP})$ as a function of the recall = $\mathrm{TP}/(\mathrm{TP}+\mathrm{FN})$. This metric is threshold-independent but in contrast to AUROC, the AUPR adjusts for the different positive and negative base rates. In our experiments, misclassifications+OOD samples are used as the positive detection class. These results confirm that KLoSNet is the best suited measure to detect simultaneously misclassifications and OOD samples.

### D.4 RESULTS WITH SVHN AS OOD TEST DATASET

We report in Fig. 10 all the results when evaluating with SVHN Netzer et al. (2011) as OOD dataset. Along with simultaneous detection results, we also provide separate results for misclassifications

Table 8: **Comparative experiments including relative Mahalanobis (Rel_Maha) and Mahalanobis (Maha) baselines on CIFAR-10 and CIFAR-100.** Misclassification (Mis.), out-of-distribution (OOD) and simultaneous (Mis+OOD) detection results (mean % AUROC and std. over 5 runs). Bold type indicates significantly best performance ($p<0.05$) according to paired t-test.

| | Method | Mis. | LSUN | | TinyImageNet | | STL-10 | |
|---|---|---|---|---|---|---|---|---|
| | | | OOD | Mis+OOD | OOD | Mis+OOD | OOD | Mis+OOD |
| **CIFAR-10** VGG-16 | Maha | 91.2 ±0.3 | **88.9** ±0.2 | **91.3** ±0.1 | **86.4** ±0.2 | 90.2 ±0.1 | 63.4 ±0.2 | 78.8 ±0.3 |
| | Rel_Maha | 91.3 ±0.4 | 88.8 ±0.2 | **91.4** ±0.2 | **86.4** ±0.2 | 90.2 ±0.1 | 63.3 ±0.2 | 78.8 ±0.2 |
| | KLoSNet | **92.5** ±0.6 | 87.6 ±0.9 | **91.7** ±0.9 | **86.6** ±0.9 | **91.2** ±0.8 | **67.7** ±1.4 | **81.8** ±0.9 |
| **CIFAR-100** VGG-16 | Maha | 84.0 ±0.2 | **71.1** ±1.0 | 82.4 ±0.5 | **77.0** ±0.5 | 84.9 ±0.3 | **75.4** ±0.3 | 84.3 ±0.5 |
| | Rel_Maha | 83.5 ±0.4 | 70.5 ±1.0 | 81.9 ±0.6 | **77.0** ±0.5 | 84.6 ±0.4 | **75.4** ±0.3 | 84.1 ±0.5 |
| | KLoSNet | **86.7** ±0.4 | 68.4 ±1.1 | **83.0** ±0.6 | 76.4 ±0.4 | **86.4** ±0.4 | 75.0 ±0.5 | **86.0** ±0.4 |

detection and OOD detection respectively. Similarly to the comparative results in the main paper, KLoSNet outperforms all the baselines in every simultaneous detection benchmark, with Mahalanobis being second.

### D.5 COMPARISON WITH RELATIVE MAHALANOBIS

In Table 8, we include results with the relative Mahalanobis distance proposed in Ren et al. (2021).

Results show no improvement with the simultaneous detection performances, and even a slight decrease for the task of misclassification detection. In the original workshop paper, the authors evaluate their method with near-OOD samples, such as CIFAR-10 vs. CIFAR-100, while we evaluate here with far OOD datasets e.g. TinyImageNet.

### D.6 IMPACT OF ADVERSARIAL PERTURBATIONS

In the original papers, ODIN and Mahalanobis preprocess inputs by adding small inverse adversarial perturbations to reinforce networks in their prediction; this has also the observed benefit to make in-distribution and out-of-distribution samples more separable. The tuning of the adversarial noise's magnitude depends on the evaluated OOD data. In Figure 11a, we plot the AUC of each detection task with different values of perturbation magnitude $\varepsilon$ with ODIN, Mahalanobis and KLoS, using SVHN as OOD dataset. Even though there exists a particular noise value for improved OOD detection (Fig. 11a, middle), increasing noise magnitude deteriorates performances in misclassification detection (Fig. 11a, left) for each method. The best results on the simultaneous detection task (Fig. 11a, right) correspond to $\varepsilon = 0$, as done in experiments presented in the main paper.

Worse, except with SVHN, adversarial perturbations are detrimental even to OOD detection. We report the AUC results of varying adversarial perturbations on CIFAR-10 dataset when using LSUN (Fig. 11b), TinyImageNet (Fig. 11c) and STL-10 (Fig. 11d) as OOD datasets. The best results on each considered task correspond to $\varepsilon = 0$ and KLoS outperforms both Mahalanobis and ODIN. As opposed to results with SVHN as OOD dataset, we did not observe improvements on any method (ODIN, Mahalanobis and KLoS) when using inverse adversarial perturbations for OOD detection with LSUN, TinyImageNet and STL-10 datasets. Similar results are observed in Liang et al. (2018) (Appendix B, Fig. 8) when using WideResNet architectures. Regarding Mahalanobis in Lee et al. (2018), the authors only provided ablation for SVHN dataset.

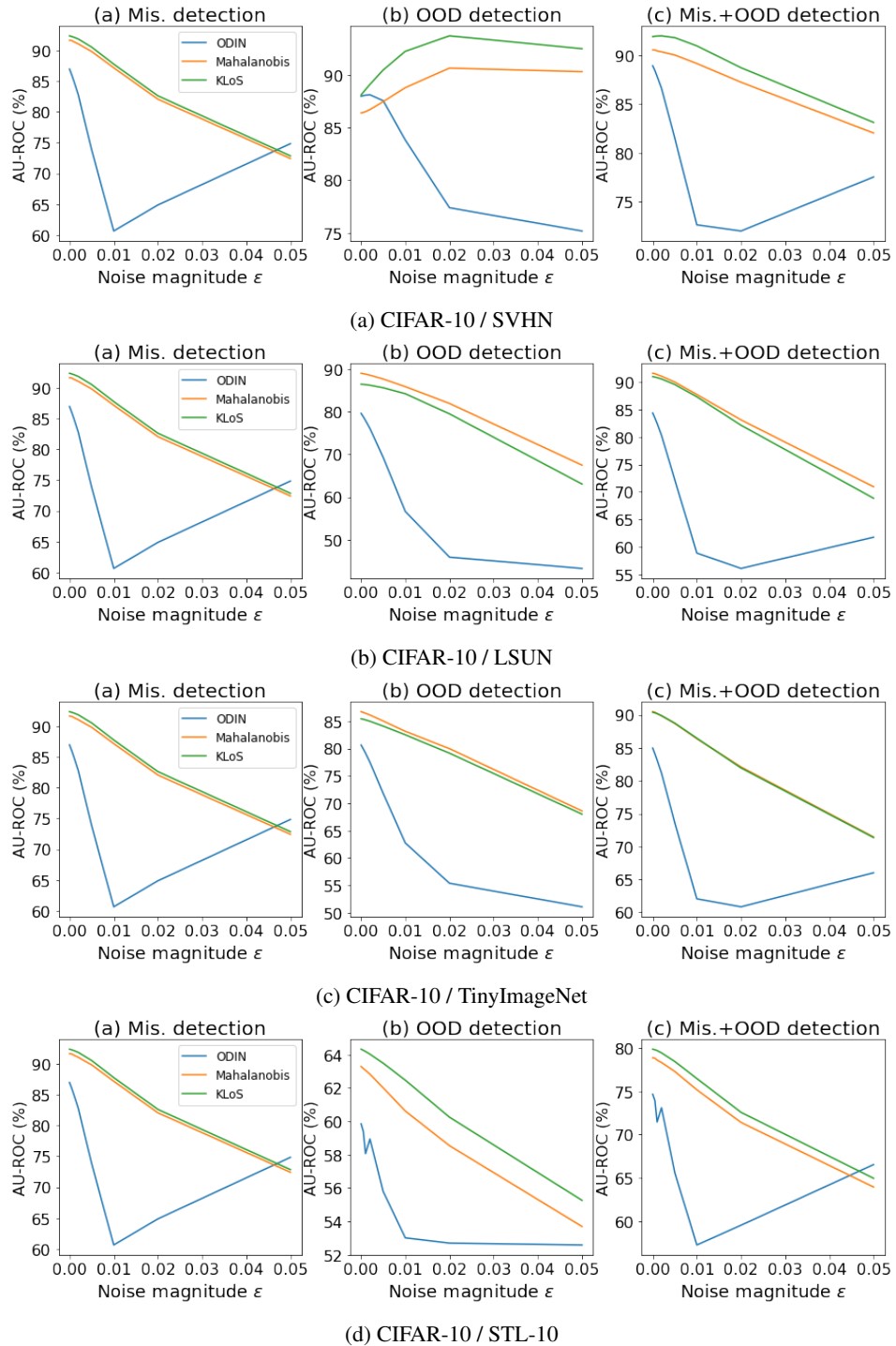

Figure 11: Effect of inverse adversarial perturbations on OOD-designed measures and KLoS for misclassification detection, OOD detection and simultaneous detection with VGG-16 architecture.

# E    SELECTIVE CLASSIFICATION IN PRESENCE OF DISTRIBUTION SHIFTS

Classification with a reject option, also known as *selective classification* El-Yaniv & Wiener (2010), consists in a scenario where a classifier can abstain on points where its confidence is below a certain threshold. This is appropriate for applications where the system can hand over to human experts or users. Performance can be measured on *risk-coverage* curves. The *coverage* is the probability mass of the non-rejected region in $\mathcal{X}$ and can be empirically estimated by the percentage of the non-rejected samples. The *risk* of a selective classifier is the average loss on the accepted samples. Given a chosen coverage, good selective classifiers correlate with low risk. Averaged performances are evaluated on risk-coverage curves with a threshold-independent area-under-curve metric, denoted here AURC. The lower the AURC, the better the selective classifier.

Previous works evaluate the performance on in-distribution data. However, a classifier may encounter data drawn from a different distribution when deployed in the wild. Following Koh et al. (2020), we extend selective classification by penalizing non-rejected OOD samples. If a sample is drawn from the in-distribution, we compute the 0/1 cost function as usual. For OOD samples, we apply the maximum cost of 1, whatever the prediction. As for simultaneous detection, we rely on oversampling to mitigate the unbalance between misclassifications and OOD samples.

Experiments are conducted with previously trained VGG-16 networks on CIFAR-100. We measure their selective classification when subject to distribution shifts by considering CIFAR-100C Hendrycks & Dietterich (2019) as OOD dataset. This dataset is constructed by corrupting the original CIFAR-100 test set. There is a total of 15 types of corruptions, which can be grouped into five families, namely *noise*, *blur*, *weather* and *digital*. Each corruption comes with five different levels of severity. While this dataset is commonly used to measure robustness to distribution shift, we focus here on models' ability to reject these samples along with misclassifications made on the original CIFAR-100 test set.

The results are reported by corruption family (noise, blur, weather and digital) in Fig. 13a and detailed in Tables 13b and 12b. One common observation regardless of the criterion is that selective classification is harder when subject to noise perturbations than other types of perturbation. In each case, KLoSNet and ConfidNet obtain the best performances. For instance, for weather perturbations on CIFAR-10-C, KloSNet achieves 42.7% AURC and ConfidNet 43.4% AURC. In particular, KloSNet outperforms every other method for blur, weather and digital perturbations of CIFAR-100-C. Hence, when subject to an unforeseen distribution shift, a model equipped with KLoSNet provides more accurate uncertainty estimates without sacrificing predictive performances. Note that for noise corruptions, the results depend widely on the run, which makes interpretation more difficult.

We present the results of a similar experiment on CIFAR10-C in Figure 12a and its detailed results in Table 12b. Here, KLoSNet obtains state-of-the-art results in every corruption and also on average with 41.0% AURC.

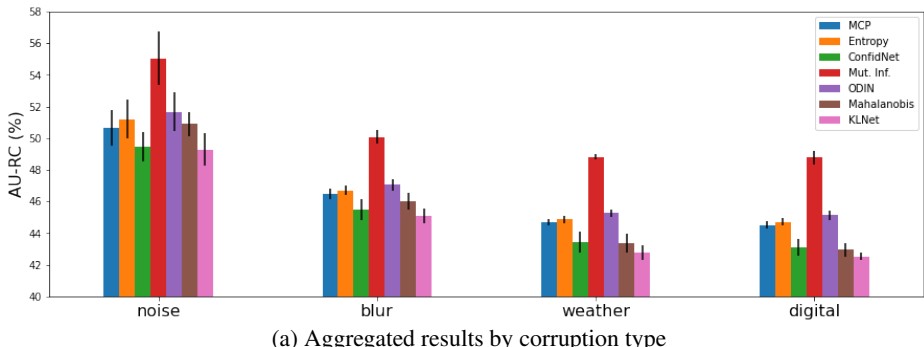

(a) Aggregated results by corruption type

| Method | Clean | Noise | | | Blur | | | | Weather | | | | Digital | | | | Mean |
|---|---|---|---|---|---|---|---|---|---|---|---|---|---|---|---|---|---|
| | | Gaussian | Shot | Impulse | Defocus | Glass | Motion | Zoom | Snow | Frost | Fog | Bright | Contrast | Elastic | Pixel | JPEG | |
| MCP | 48.3% | 46.7% | 48.0% | 43.7% | 48.6% | 45.7% | 45.4% | 43.4% | 44.4% | 42.5% | 40.4% | 46.5% | 43.3% | 43.3% | 43.0% | 1.9% | 42.2% |
| Entropy | 48.8% | 47.2% | 48.5% | 43.9% | 49.1% | 45.9% | 45.6% | 43.6% | 44.7% | 42.7% | 40.6% | 46.9% | 43.5% | 43.7% | 43.2% | 2.0% | 42.5% |
| ConfidNet | 47.4% | 45.8% | 47.1% | 42.9% | 48.1% | 45.2% | 44.9% | 42.5% | 43.5% | 41.6% | 39.1% | 45.9% | 42.6% | 42.0% | 42.1% | 1.3% | 41.4% |
| Mut. Inf. | 53.6% | 52.3% | 53.2% | 48.3% | 53.0% | 49.3% | 49.1% | 48.8% | 49.8% | 47.8% | 46.9% | 51.5% | 47.9% | 49.8% | 48.2% | 4.8% | 47.1% |
| ODIN | 49.3% | 47.7% | 48.9% | 44.3% | 49.5% | 46.3% | 46.0% | 44.1% | 45.2% | 43.2% | 41.2% | 47.3% | 43.9% | 44.3% | 43.7% | 2.2% | 42.9% |
| Mahalanobis | 48.9% | 46.9% | 48.6% | 42.5% | 49.7% | 45.3% | 44.8% | 43.1% | 44.1% | 41.4% | 38.6% | 45.8% | 42.6% | 42.9% | 42.4% | 1.0% | 41.8% |
| KLoSNet | 47.0% | 45.3% | 46.9% | 42.3% | 48.0% | 45.0% | 44.6% | 42.2% | 43.1% | 41.1% | 38.1% | 45.2% | 42.4% | 41.8% | 42.0% | 0.9% | 41.0% |

(b) Detailed results

Figure 12: **Selective classification on CIFAR10-C**. Comparative performance in AURC (%) of classification with the option to reject misclassified test samples and samples from shifted distributions. Results are average on 5 runs (mean ± std.).

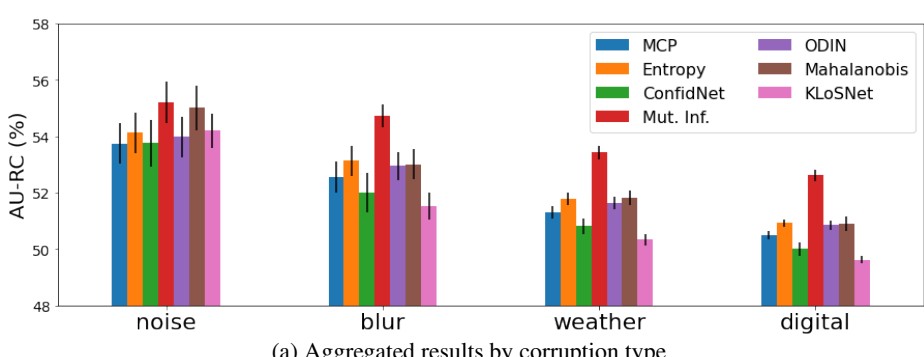

(a) Aggregated results by corruption type

| Method | Clean | Noise | | | Blur | | | | Weather | | | | Digital | | | | Mean |
|---|---|---|---|---|---|---|---|---|---|---|---|---|---|---|---|---|---|
| | | Gaussian | Shot | Impulse | Defocus | Glass | Motion | Zoom | Snow | Frost | Fog | Bright | Contrast | Elastic | Pixel | JPEG | |
| MCP | 53.3% | 52.7% | 55.2% | 50.8% | 54.0% | 52.8% | 52.5% | 51.3% | 52.0% | 50.5% | 47.9% | 52.1% | 50.7% | 50.4% | 51.3% | 13.1% | 49.4% |
| Entropy | 53.5% | 53.0% | 55.8% | 51.3% | 54.8% | 53.3% | 53.1% | 51.8% | 52.6% | 51.0% | 48.2% | 52.7% | 51.1% | 50.9% | 51.7% | 13.5% | 49.9% |
| ConfidNet | 53.5% | 52.8% | 55.0% | 50.1% | 53.7% | 52.3% | 51.9% | 50.8% | 51.6% | 50.0% | 47.2% | 51.7% | 50.3% | 49.8% | 50.9% | 11.7% | 49.0% |
| Mut. Inf. | 54.5% | 54.0% | 57.1% | 53.1% | 56.2% | 54.9% | 54.7% | 53.5% | 54.3% | 52.6% | 50.0% | 54.4% | 52.4% | 53.0% | 53.2% | 16.5% | 51.5% |
| ODIN | 53.4% | 52.9% | 55.5% | 51.2% | 54.5% | 53.2% | 52.9% | 51.7% | 52.4% | 50.9% | 48.2% | 52.5% | 51.0% | 50.8% | 51.7% | 13.7% | 49.8% |
| Mahalanobis | 54.5% | 53.9% | 56.7% | 50.9% | 55.5% | 52.8% | 52.9% | 52.2% | 52.5% | 50.7% | 47.8% | 52.4% | 51.0% | 51.4% | 51.9% | 11.0% | 49.9% |
| KLoSNet | 54.0% | 53.1% | 55.5% | 49.5% | 53.6% | 51.6% | 51.4% | 50.7% | 51.1% | 49.2% | 46.5% | 51.3% | 49.9% | 49.7% | 50.7% | 9.7% | 48.6% |

(b) Detailed results

Figure 13: **Selective classification on CIFAR100-C**. Comparative performance in AURC (%) of classification with the option to reject misclassified test samples and samples from shifted distributions. Results are average on 5 runs (mean ± std.).

