# OpenReview forum: "Effective Uncertainty Estimation with Evidential Models for Open-World Recognition"
_ICLR.cc/2022/Conference — ICLR 2022 Submitted_

### Official Review · Reviewer_rode · 2021-10-25

**Correctness:** 1
**Technical Novelty And Significance:** 2
**Empirical Novelty And Significance:** 1
**Recommendation:** 3
**Confidence:** 3

**Main Review:**

The paper provides nicely drawn illustrative figures that explain the impact of first-order and second-order uncertainties on uncertainty quantification profiles of classification models.

The paper suffers from multiple fundamental weaknesses detailed below:

i) The paper lacks a concrete hypothesis. Which particular problem of the "existing" evidential learning problem is being targeted? Figure 1 only motivates the evidential learning framework, but does not point to any weakness of it. The claim in Page 2 that the evidential models require auxiliary OOD data holds for the Prior Networks approach (Malinin & Gales, 2018, 2019) but not for EDL (Sensoy et al., 2018). As I understand, the only novelty claim is to use Eq 5 as an OOD and misclassification detection criterion instead of the well-established mutual information or entropy criteria. However, I am having hard time to decipher what the claimed limitation of the mutual information criterion is. All the argumentation in the paper is valid only against MCP, which is obvious for the uncertainty quantification community, but what is the problem with mutual information, which in essense has mechanisms for handling both in-domain and out-of-domain uncertainties? For example, the claim:

"To capture uncertainties due to class confusion and lack of evidence, an effective measure should account for the sharpness of the Dirichlet distribution and its location on the simplex."

holds also for the mutual information criterion.

ii) The only section in the paper where there is a claim about a methodological novelty is Section 2.2. I am having hard time to make sense of the criterion proposed in Eq 5. It first calculates the Dirichlet density corresponding to an input pattern that captures both first and second-order uncertainties, then makes a heuristic choice of comparing it to a spiky Dirichlet distribution. Why should it be better than decomposing the uncertainty of the predictive distribution according to the law of total variance or any other decomposition criterion that has an information-theoretic meaning? I also do not understand why divergence from the spiky (winner-take-all) version of the own prediction of a model should correlate with the probability of an input sample coming from a different domain.

iii) The paper does not report experimental results that give a decisive outcome about the central claims of the paper. As I understand, all models in Table 1 are trained using Eq 6 first, which corresponds to the Reverse KL design of Malinin et al., 2019, just without OOD data. It is then blurry what exactly happens. Is then only KLoSNet trained further on Eq 8? If so, the comparison is unfair, as the proposed solution gains more training resources than its counterparts. If all other models are also trained further with Eq 8, the comparison is still unfair, because they are trained further with a loss that is designed for another OOD criterion than they are designed for. A decisive comparison could have been as follows:

a) Take EDL loss "literally". Train it for X epochs.

b) Take Reverse KL loss "literally". Train it for X epochs.

c) Take Eq 8 and train KLoSNet "without" any pretraining for X epochs.

Evaluate all three models with respect to both mutual information and KLoS scores. Now compare which score and which model gives best OOD performance and draw the scientific claim accordingly.

The results in favor of KLoSNet reported in Table 1 may have something to do with post-training KLoSNet with respect to a criterion that is later used as a performance score. This would of course make both the trained model and the chosen criterion advantageous over others, but it would be the same if any other criterion was chosen.

**Summary Of The Paper:**

The paper studies the problem of doing out-of-domain and misclassification detection using the evidential learning framework, which suggests modelling the second-order uncertainties of class assignment probabilities with a Dirichlet-Multinomial hierarchical model. The model maps an input observation, be it an image, to the concentration parameters of the Dirichlet prior on the class probabilities by a neural net with an arbitrary architecture. On top of this existing framework (Sensoy et al., 2018, Malinin & Gales 2018), the paper proposes to use a KL divergence based decision criterion, called "KL on Simplex (KLoS)",  which fits a reverse KL divergence between the model output and a spike on the predicted class label in order to smoothen the predictions.

**Summary Of The Review:**

Overall, the paper lacks a clear hypothesis statement (Item i)), it does not propose a novel solution (Item ii)), and it does not conduct a set of experiments that give conclusive results in favor of a solid scientific hypothesis (Item iii)). Hence my initial grade is a clear reject.

---

> ### Author Response · Authors · 2021-11-17
> **Response to Reviewer rode (1/2)**
>
> We thank you for your comments. However we think that the review contains several confusions about our work and we address them in detail below.
>
> **“The paper lacks a concrete hypothesis. Which particular problem of the "existing" evidential learning problem is being targeted?”**
>
> In this paper, we show that the current uncertainty measures used with evidential models are not suited for the task of simultaneous detection of misclassifications and OOD samples. First, in Figure 1, we illustrate the limits of the first-order uncertainty measures, such as MCP and predictive entropy, which are invariant to the spread of the Dirichlet distribution as they reduce probability distribution on the simplex to their expected values (probabilities). Then, we also show why second-order measures such as EPKL and mutual information are not efficient when used on models trained without OOD data. Indeed, the rely on the assumption that all OOD samples will have a lower precision \alpha_0 than in-distribution samples, an assumption not fulfilled for every OOD samples and shown empirically in Figure 2.
>
> **“Figure 1 only motivates the evidential learning framework, but does not point to any weakness of it. ”**
>
> Figure 1 illustrates the limits of the first-order uncertainty measures, such as MCP and predictive entropy. The caption and the last paragraph of page 1 contains a detailed explanation.
>
> **“The claim in Page 2 that the evidential models require auxiliary OOD data holds for the Prior Networks approach (Malinin & Gales, 2018, 2019) but not for EDL (Sensoy et al., 2018).”**
>
> While the original EDL does not use OOD training data, following work of [Sensoy et al., 2020] propose to synthesize outliers with a generative adversarial network to overcome the use of OOD training data.
> [Sensoy et al., 2020] Uncertainty-Aware Deep Classifiers using Generative Models, AAAI 2020
>
> **“All the argumentation in the paper is valid only against MCP, which is obvious for the uncertainty quantification community”**
>
> The claim developed in Figure 1 is valid for all uncertainty measures that rely only on the first-order probabilities, including MCP and predictive entropy.
>
> **“what is the problem with mutual information, which in essense has mechanisms for handling both in-domain and out-of-domain uncertainties?”**
>
> Existing second-order uncertainty measures such as EPKL and mutual information rely on the assumption that all OOD samples will have a lower precision \alpha_0 than in-distribution samples. As stated in the introduction and acknowledged in recent works [Charpentier et al. 2020, Sensoy et al. 2020], this assumption does not hold when training without OOD training data. This behaviour is also verified in our image experiments (Figure 2) and in the synthetic experiment (Figure 5). In contrast, KLoS acts as a class-wise density estimator, which enables a much more robust OOD detection when no OOD training data are available.
>
> **“The only section in the paper where there is a claim about a methodological novelty is Section 2.2”**
>
> Our work introduces three contributions in the introduction, including two methodological claim: (1) the use of KLoS as an uncertainty measure for simultaneous detection , and (2) a confidence learning scheme based on a refined KLoS objective to further improve performance by using an auxiliary confidence network.
>
> **“Why should it be better than decomposing the uncertainty of the predictive distribution according to the law of total variance or any other decomposition criterion”**
>
> The decomposition of the predictive entropy proposed in [Malinin et al., 2018, Eq. 16] makes the assumption that predictive entropy corresponds to an estimate of the total uncertainty. However, as shown in the illustrated example in Figure 1, entropy only captures uncertainty due to class confusion. A finding also shared in non-evidential paper such as DUQ [van Amersfoort et al., NeurIPS 2020] and DDU [Mukhoti et al., 2021], mentioned by reviewer oef4.

---

> > ### Author Response · Authors · 2021-11-17
> > **Response to Reviewer rode (2/2)**
> >
> > **“I also do not understand why divergence from the spiky (winner-take-all) version of the own prediction of a model should correlate with the probability of an input sample coming from a different domain.”**
> > - KLoS captures both uncertainty and can be decomposed into two terms with Eq. (6). In Appendix B, we show that the negative log-likelihood term relates to class confusion while the remainder helps to detect OOD samples, as illustrated on the synthetic data in Figure 9.
> > - Another argument in favor of KLoS is that it naturally reflects the loss used in training by design. Hence, by mimicking the evidential training objective, we reflect the fact that the model is confident about its prediction if KLoS is close to zero (page 5)
> >
> > **“Is then only KLoSNet trained further on Eq 8? If so, the comparison is unfair, as the proposed solution gains more training resources than its counterparts.”**
> > - During confidence learning (Eq. 8), classifier weights are always fixed. The goal of confidence learning is to improve the uncertainty estimation of errors on a fixed classifier.  Hence, the training of the classifier and the training of the confidence network are different, the latter being initialized with the weights of the former to help convergence.
> > - All the uncertainty measures are taken from an evidential model trained with Eq. 6. KLoSNet’s results correspond to the output of the confidence network, whose training is based on the classifier’s output.
> > - By comparing KLoS’ results (Table 2) with baselines in Table 1, the ablation study shows that KLoS is already improving detection performance over current baselines

---

> > > ### Comment · Reviewer_rode · 2021-11-29
> > > **Keep my score**
> > >
> > > Thanks for your response. I do not think I received a clear enough yes/no answer to my question regarding the fairness of the comparison across models. For one reason or another, if KLoS has access to more resources, then the comparison is not valid. The decisive comparison I suggested in my original review is not fulfilled.
> > >
> > > Methodological novelty: The author response confirms my statement.
> > >
> > > Fig 1 talks about the limits of first-order uncertainty measures but not the limits of EDL per se. The whole point of EDL is learning second-order uncertainties.
> > >
> > > The response to my comment "“what is the problem with mutual information, which in essense has mechanisms for handling both in-domain and out-of-domain uncertainties?” is not correct. OOD samples by definition need to have lower precision. If they do not, it is because the model's uncertainty predictions are inaccurate. Then, the way to go is to repair the predictions, not the evaluation metric.

---

> > > > ### Author Response · Authors · 2021-11-29
> > > > **Extended discussion with Reviewer rode**
> > > >
> > > > Dear Reviewer rode, we thank you for your feedback.
> > > >
> > > > - Regarding resources comparison, we respectfully disagree with your comment: during confidence training, computational resources are used to learn the confidence network, which is separate from the classifier. In addition, the ablation study shows that the criterion KLoS is already improving detection performance over current baselines, without using any further resources. We hope our response addresses your concern in the same way as it did for Reviewer FYgq.
> > > >
> > > > -  While EDL provide a second-order uncertainty representation, we showed in Figure 2 that without using OOD training data, unseen OOD samples can have larger precision $\alpha_0$ than ID samples. This behavior explain why second-uncertainty measures only accounting for the spread of the Dirichlet distribution on the simplex can fail to detect some OOD samples. Hence, along with Figure 1, we showed that current uncertainty measures with evidential models are not suited to the task of simultaneous detection in this setting.
> > > >
> > > > - "OOD samples by definition need to have lower precision". => Our work, along with those of [Charpentier et al. 2020] and [Sensoy et al. 2020], showed that in practice this is not what we observed when training without OOD training data. On a more general mater, finding suitable OOD training samples may be easy for some academic datasets but can be more problematic
> > > > in real-world applications, with the risk of degrading performance with an inappropriate choice. This is shown in Figure 8 and detailed in Section 4.3: the choice of the OOD training dataset significantly impact the performances of OOD detection. In contrast, our proposed measure KLoS is more robust to this choice. To the best of our knowledge, this is the first work to study the impact of the choice of OOD training dataset and we hope this will help the community to focus more on how to build suitable OOD training sets, in line of [Sensoy et al. 2020].

---

### Official Review · Reviewer_oef3 · 2021-10-26

**Correctness:** 2
**Technical Novelty And Significance:** 2
**Empirical Novelty And Significance:** 3
**Recommendation:** 5
**Confidence:** 4

**Main Review:**

The \lambda term seems to be an important hyper-parameter but it was unclear (unless I missed it!) how it was tuned or how sensitive it is.

It would have been helpful to frame the proposal in terms of standard notions of epistemic (model) and aleatoric (irreducible) uncertainty. If the proposed measure is meant to capture epistemic uncertainty, would we expect this value to be greater on average on held-out data, for smaller training datasets? If so, it'd be nice to show that via experiments training with different amounts of data and showing this value exhibits the desired trends.

Overall, while the experiments suggest that the proposed method is doing something reasonable, it's not clear to me that it's doing something reasonable for the purported reasons. Besides the other suggestions above, some additional model ablations may be helpful to demonstrate the benefits of the proposed approach, such as auxiliary regression of simplified versions of the proposed objective.

Separate from my concerns about the justification for the method, I also found there to be a lack of references to related recent work and I think the paper would be more convincing with consideration of stronger baselines. Experimental comparisons aren't strictly necessary in all cases, but at minimum further discussion would be help position this work relative to other deterministic approaches, such as:

- The Mahalanobis baseline is competitive with the proposed approach, but more recent, improved versions of Mahalanobis have been proposed for OOD: https://arxiv.org/pdf/2106.09022.pdf
- Deep deterministic uncertainty (DDU): https://arxiv.org/pdf/2102.11582.pdf
- MIR, see: https://arxiv.org/pdf/2107.00649.pdf
- Deterministic uncertainty quantification (DUQ): https://arxiv.org/pdf/2003.02037.pdf



**Summary Of The Paper:**

This paper suggests training an auxiliary DNN to predict the uncertainty associated with a given example. The auxiliary model uses the predictions of a model fit to the predict the concentration parameters of a Dirichlet (an "evidential neural network") to measure the KL divergence between the predicted posterior and one based on the ground truth labels. (The auxiliary model is necessary because the ground truth labels are not available at test time.) The primary claim is that "[...] leveraging the second-order uncertainty representation that evidential models provide, KLoS captures both class confusion and lack of evidence in a single score." In a series of experiments, it is found that the proposed model is competitive with or outperforms selected alternate models across several benchmarks for OOD detection and misclassification detection.

**Summary Of The Review:**

Overall, this paper proposes an interesting and to my knowledge novel approach to uncertainty quantification. However, I believe it has two main weaknesses: (1) there's a lack of discussion and experimental comparisons to recent work in uncertainty estimation and (2) there isn't enough introspection of the proposed approach to confirm it's working for the purported reasons.

---

> ### Author Response · Authors · 2021-11-17
> **Response to Reviewer oef3**
>
> We thank you for your meaningful and valuable comments. Our answers to your questions are available below.
>
> &nbsp;
>
> **[Experiment with held-out data]**
>
> We thank the reviewer for this interesting experiment suggestion. Our current experiments showed that KLoS value is greater for OOD samples, which is also linked to epistemic uncertainty.
>
> &nbsp;
>
> **[Ablation on the confidence learning module]**
>
> In Table 2, we provided an ablation study to evaluate the impact of the confidence learning module. The comparison with ConfidNet baseline in Table 1 also show the impact of the criterion choice as ConfidNet’s learning is based on the true class probability criterion while KLoSNet learned to regress KLoS* criterion. Finally, we provide intermediate results in the following table before fine-tuning, which corresponds to training the confidence head only (KLosNet-sh).
>
> |                 |              |              |     **LSUN**     |              | **TinyImageNet** |              |    **STL-10**    |              |
> |-----------------|--------------|:------------:|:------------:|:------------:|:------------:|:------------:|:------------:|:------------:|
> |                 | **Method**   |     Mis.     |      OOD     |   Mis.+OOD   |      OOD     |   Mis.+OOD   |      OOD     |   Mis.+OOD   |
> |  **CIFAR-10**   | KLosNet-sh   | 90.7 (+-0.6) | 86.2 (+-1.7) | 90.3 (+-0.7) | 85.7 (+-0.5) | 90.1 (+-0.4) | 65.2 (+-0.3) | 79.7 (+-0.4) |
> |     VGG-16      | KLosNet-ft   | **92.5** (+-0.6) | **87.6** (+-0.9) | **91.7** (+-0.9) | **86.6** (+-0.9) | **91.2** (+-0.8) | **67.7** (+-1.4) | 81.8 (+-0.9) |
> | **CIFAR-100**   | KLosNet-sh   | 84.8 (+-0.5) | 67.9 (+-1.1) | 81.3 (+-0.6) | 75.5 (+-0.5) | 84.6 (+-0.5) | 73.7 (+-0.6) | 83.8 (+-0.5) |
> |     VGG-16      | KLosNet-ft   | **86.7** (+-0.4) | **68.4** (+-1.1) | **83.0** (+-0.6) | **76.4** (+-0.4) | **86.4** (+-0.4) | **75.0** (+-0.5) | 86.0 (+-0.4) |
>
> We observe that fine-tuning KLoSNet’s encoder brings further improvements in every detection task.
>
> &nbsp;
>
> **[Further baselines]**
>
> We thank the reviewer for pointing out these papers that we will include in our related work.
> - We added results with Relative Mahalanobis distance on CIFAR-10 and CIFAR-100 datasets with VGG-16 networks in the following table.
>
> |                 |              |              |     **LSUN**     |              | **TinyImageNet** |              |    **STL-10**    |              |
> |-----------------|--------------|:------------:|:------------:|:------------:|:------------:|:------------:|:------------:|:------------:|
> |                 | **Method**   |     Mis.     |      OOD     |   Mis.+OOD   |      OOD     |   Mis.+OOD   |      OOD     |   Mis.+OOD   |
> |  **CIFAR-10**   | Maha  | 91.2 (+-0.3) | **88.9** (+-0.2) | **91.3** (+-0.1) | **86.4** (+-0.2) | 90.2 (+-0.1) | 63.4 (+-0.2) | 78.8 (+-0.3) |
> |      VGG-16     | Rel_Maha  | 91.3 (+-0.4) | **88.8** (+-0.2) | **91.4** (+-0.2) | **86.4** (+-0.2) | 90.2 (+-0.1) | 63.3 (+-0.2) | 78.8 (+-0.2) |
> |                 | KLosNet   | **92.5** (+-0.6) | 87.6 (+-0.9) | **91.7** (+-0.9) | **86.6** (+-0.9) | **91.2** (+-0.8) | **67.7** (+-1.4) | 81.8 (+-0.9) |
> | **CIFAR-100**   | Maha   | 84.0 (+-0.2) | **71.1** (+-1.0) | 82.4 (+-0.5) | **77.0** (+-0.5) | 84.9 (+-0.3) | **75.4** (+-0.3) | 84.3 (+-0.5) |
> |      VGG-16     | Rel_Maha   | 83.5 (+-0.4) | **70.5** (+-1.0) | 81.9 (+-0.6) | **77.0** (+-0.5) | 84.6 (+-0.4) | **75.4** (+-0.3) | 84.1 (+-0.5) |
> |                 | KLosNet   | **86.7** (+-0.4) | 68.4 (+-1.1) | **83.0** (+-0.6) | **76.4** (+-0.4) | **86.4** (+-0.4) | **75.0** (+-0.5) | **86.0** (+-0.4) |
>
> Results show no improvement in the simultaneous metric, and even a performance decrease for the task of misclassification detection. In the original workshop paper, the authors evaluate their method on near-OOD samples, such as CIFAR-10 vs. CIFAR-100, while we evaluate here with far OOD datasets e.g.TinyImageNet.
>
> - DDU and MIR also used density estimates from hidden representations to estimate epistemic uncertainty and predictive entropy to estimate aleatoric uncertainty. With KLoS, we propose a principled approach to estimate simultaneously aleatoric and epistemic uncertainty;
> - DUQ shares our finding that predictive entropy only captures aleatoric uncertainty . DUQ also considers class-based densities but in contrast with the decomposition of KLoS (Eq. 6), there is no formal way to distinguish between the sources of uncertainty with DUQ (Section 2.3).

---

### Official Review · Reviewer_ucNF · 2021-11-02

**Correctness:** 3
**Technical Novelty And Significance:** 3
**Empirical Novelty And Significance:** 2
**Recommendation:** 5
**Confidence:** 5

**Main Review:**

+ves:

+ The idea of using a class-wise divergence measure based on evidential models is interesting.

+ The paper has a nice presentation overall. The figures are effective in conveying the key idea and intuition.

+ The results section is well structured. It's nice to see both OOD detection and ID misclassification performance.

Concerns:

- Authors highlight that the uncertainty estimation should consider both class confusion and lack of evidence. However, this may not be always desirable for OOD uncertainty estimation. Take an extreme example, in the classification task of "bird" vs "car", given images with "A: bird sitting on the car" and "B: computer", the model would have more class confusion in the former case. Yet A is an in-distribution image and B is an OOD image.

- The above concern brings to my next question: have the authors investigated the efficacy of the two individual terms in Equation (6) for OOD uncertainty estimation?

- "KLoS also penalizes samples having a different precision α_0. " I understand that lower precision should be penalized but why a higher amount of evidence is not preferred?

- The experimental results are based on relatively small model capacity (ResNet-18 at the largest). It is unclear how well the method scale for common capacity (say ResNet-101). This is particularly concerning, as Mahalanobis (the best baseline considered) is relatively disadvantaged in this small-capacity setting where the representation might not be as optimal as larger models. Even in this setting, the results are not conclusively in favor of the proposed method, and only is marginally better than Mahalanobis. With that being said, the results would have been more complete if a consistent trend is shown at a larger model capacity.

- Missing  OOD detection baselines Gram Matrix [1],  Generalized ODIN [2].

- Lastly, it'd be great if the authors can comment on the low performance of common baselines in Table 1. For example, the reviewer uses a standard ResNet-18 trained on CIFAR-100, using the same learning rate and schedule as detailed on Page 16. The OOD uncertainty estimation performance (on LSUN) is not matchable to Table 1. For example, using MCP yields AUROC 75.73, using ODIN and Mahalanobis yields 82.13 and 82.98 respectively. In contrast, the performance for the baselines methods seems low across the board (~70-75%).


References

[1] C. Sastry, S. Oore. Detecting Out-of-Distribution Examples with In-distribution Examples and Gram Matrices. ICML 2020.

[2] Y. Hsu, Y. Shen, H. Jin, Z. Kira. Generalized ODIN: Detecting Out-of-Distribution Image Without Learning From Out-of-Distribution Data. CVPR 2020.


**Summary Of The Paper:**

This paper proposes an uncertainty measurement KLoS by computing the KL divergence between the Dirichlet distribution from the Evidential Neural Networks and a class-wise prototype concentrated on the predicted class. An extra network KLoSNet is further trained to align with the evidential training objective. Empirical studies are performed to show the efficacy of both OOD uncertainty estimation and ID misclassification detection. Results are shown on the task of (1) synthetic dataset with three Gaussian-distributed classes and (2) CIFAR datasets.



**Summary Of The Review:**

In summary, this paper presents an interesting idea however the experimental evaluations are not sufficiently convincing in the current state. I would like to hear the authors address my concerns raised above.

---

> ### Author Response · Authors · 2021-11-17
> **Response to Reviewer ucNF**
>
> We thank you for your detailed feedback, and address your concerns below.
>
> &nbsp;
>
> **[Motivation of simultaneous detection]**
>
> The goal of our paper is to develop an uncertainty measure that encompasses both types of uncertainty to ensure safe monitoring for real-world image recognition. In this task, a good measure should enable us to distinguish between errors/OOD samples and correct predictions. In your example, this corresponds to comparing the uncertainty estimates of A and B to a low class-confusion in-distribution sample, such as “C: a bird predicted as bird by the model”.
>
> &nbsp;
>
> **[Ability of KLoS to detect misclassification and OOD samples separately]**
>
> We present quantitative results and illustrated for the toy dataset in Figure 9 of the submission (Appendix B). We observe that the NLL term, which is equivalent to MCP measure, helps to detect misclassifications while the second term denotes increasing uncertainty as we move either far away from training data. Hence, by using either the NLL term or the second term, one could distinguish the source of uncertainty if needed.
>
> &nbsp;
>
> **[Why penalize high precision samples]**
>
> Existing second-order uncertainty measures such as EPKL and mutual information rely on the assumption that all OOD samples will have a lower precision \alpha_0 than in-distribution samples. As stated in the introduction and acknowledged in recent works [Charpentier et al. 2020, Sensoy et al. 2020], this assumption does not hold when training without OOD training data. This behaviour is also verified in our image experiments (Figure 2) and in the synthetic experiment (Figure 5). In contrast, KLoS acts as a class-wise density estimator, which enables it to penalize both higher and lower precision than the in-distribution target.
>
> &nbsp;
>
> **[Experiments with large models]**
> - In line with previous papers evaluating evidential models [Malinin et al. 2018,2019; Charpentier et al. 2020; Joo et al. 2020, Nandy et al. 2020], we report performances with a VGG16 backbone.
> - We included experiments with a ResNet-18 which show similar trends with improved scores in all settings. We also took care to evaluate our method for various in-distribution datasets (CIFAR-10 and CIFAR-100) and OOD datasets (TinyImageNet, LSUN, STL-10 and SVHN).
> - The main paper and the appendix relate an intensive set of experiments. For instance, in Appendix E, we also pursued complementary experiments to evaluate our method for the task of selective classification in presence of distribution shifts with CIFAR-10C and CIFAR-100C datasets. Finally, Section 4.3 presents novel results on the impact of the OOD dataset used in training, which hasn’t been evaluated in previous papers and may provide interesting insights for the community.
>
> &nbsp;
>
> **[Comment on experimental results]**
> - Our evidential model is similar to the Prior Network with reversed KL-divergence (PN-RKL) introduced by Malinin et al. (2019). We took care in our experiments to have baseline results in line with those available in the literature, such as Nandy et al. (2020) (‘DPNRev’ in Table 2 of their paper). For instance, when using CIFAR-100 as OOD training data, they report 89.6% AUC-Mis. with MCP on CIFAR-10 and we obtain 91.1% in a similar setting (Section 4.3, Fig. 6). We even obtained better results for Mut-Inf on CIFAR-10: 88.7% vs. 91.5% in our experiments.
> - Regarding OOD uncertainty estimation performance, our results are lower in our setting as we don’t use OOD training data. Nevertheless, in Figure 8, our model trained with CIFAR-10 as in-distribution dataset and CIFAR-100 as OOD training data obtains similar results (~92% AUROC) than in previous papers.

---

> > ### Comment · Reviewer_ucNF · 2021-11-29
> > **thank you for the response**
> >
> > Thank you for the response. After reading the author's response and other reviews, I find the work not convincing for ICLR at its stage.
> >
> > I remain concerned about the experiments regarding:
> > - (1) the significance of the results w.r.t competitive OOD detection baselines and whether baselines are properly reproduced. The numbers are off by a non-trivial margin even compared to the setting without using OOD data (i.e., directly estimating the class-conditional Gaussians on the penultimate layer).
> > - (2) It is also unclear whether the method scale beyond ResNet-18 and VGG and whether it can outperform in those settings. As mentioned, the smaller model also gives an unfair disadvantage to Mahalanobis.
> > - (3) Missing more recent OOD detection baselines.

---

> > > ### Author Response · Authors · 2021-11-29
> > > **Extended discussion with Reviewer ucNF**
> > >
> > > Dear Reviewer ucNF, we thank you for reading our response and for your feedback. To answer to your remaining concerns,
> > >
> > > 1.  We'd like to point out that our baselines are in line with those reported in the literature [Malinin et al. 2019, Nandy et al. 2020] for an evidential model trained without OOD training data;
> > > 2.  In our experiments, Mahalanobis is the second-best method when used with evidential models, which reflects its good behaviour. Our experimental setting already extend previous works on evidential models by using ResNet-18 architecture. Larger models will certainly help to obtain better uncertainty performances for Mahalanobis but we see no reason why it could not be the case for KLoS as well.
> > > 3. In our work, we took care to include many relevant uncertainty measures commonly used with evidential models and more general OOD detection methods such as ODIN and Mahalanobis. We would be happy to include the two baselines mentioned by Reviewer ucNF in a revised version.
> > >
> > > We agree that reproducibility is important to truly enable the community to assess the quality of proposed methods. This is why we plan to release the code if accepted.

---

### Official Review · Reviewer_FYgq · 2021-11-02

**Correctness:** 3
**Technical Novelty And Significance:** 2
**Empirical Novelty And Significance:** 3
**Recommendation:** 6
**Confidence:** 3

**Details Of Ethics Concerns:**

I have no ethics concerns.

**Main Review:**

The paper shows good results and an extensive evaluation on various datasets and two (smaller) network architectures. Additionally, a toy dataset showcases the differences between typical measures and the proposed KLoS well.
There are three main weaknesses:

#### W1 - Baseline Comparisons
To my understanding from the appendix, KLoSNet is trained in three stages. First a standard classifier is trained based on (Joo et al., 2020) which corresponds to Eq 6. Then, the weights are frozen and a 5 layer MLP is trained to predict KLoS* from the final layer of the original network. Finally, the whole network is finetuned.
This procedure makes it difficult to compare the performance in Tab. 1, as the baselines do not receive the extra training, which potentially conflates the performance gain through longer training with the benefits of the proposed method.

#### W2 - Joint Metric
KLoS is designed as a joint metric for classification confidence and OOD detection. This requires combining the two concepts into one score by implicitly weighing them. It is not clear that this step is meaningful or desired. It might be beneficial to know whether a prediction is uncertain because of class uncertainty or because the sample is far from the distribution.

#### W3 - Ablation
While some hyperparameters such as the oversampling rate $\kappa$ are ablated, some important experiments are missing:

The choice of the prototype Dirichlet distribution $\gamma_{y}$ is potentially an important hyper parameter. It would be interesting to understand the influence of the choice of precision $\alpha$ as this controls the ‘smoothing’ of the actual target.

The proposed training scheme relies on initialization of KLoSNet with the original classifier. Is this important to learn a proper measure or does this simply speed up convergence?

#### W4 -  Minor Concerns
Secon 4.1 states that the Mahalanobis metric ``may have trouble detecting misclassifications’’, however the results in Tab.1 do not seem to confirm this explanation.


**Summary Of The Paper:**

The paper proposes a method for uncertainty estimation of a classifier network called “KLoSNet”. The KLoS metric is designed to measure both the uncertainty between classes and the probability of out-of-domain samples. For a given sample, the estimated KLoS value is predicted from a secondary network that is trained to predict the true KLoS score of the first since ground truth class information is not available during test time.
The method is evaluated on a standard set of benchmarks and shows good performance with respect to recent work.

**Summary Of The Review:**

The design of the evaluation makes it difficult to understand exactly where the performance improvement comes from (W1) as the proposed method seems to be trained ~50% longer than the baselines. Additionally, I am not fully convinced that a joint metric for OOD and misclassification is desired (W2). Finally, some ablation experiments are needed to understand the effect of certain design choices (W3).

---

> ### Author Response · Authors · 2021-11-17
> **Response to Reviewer FYgq**
>
> We thank you for your detailed comments. Our answers to your questions are available below.
>
> &nbsp;
>
> **W1 - Baseline Comparisons**
> - During confidence learning (step 2 & 3), classifier weights are always fixed. The goal of confidence learning is to improve the uncertainty estimation of errors on a fixed classifier.  Hence, the training of the classifier and the training of the confidence network are different, the latter being initialized with the weights of the former to help convergence.
> - By comparing KLoS’ results (Table 2) with baselines in Table 1 (i.e. without using the confidence learning module), the ablation study shows that KLoS is already improving detection performance over current baselines
>
> &nbsp;
>
> **W2 - Joint Metric**
> - When deployed in open-world conditions, a model should be able to discard both in-distribution misclassifications and OOD samples to ensure safety. Hence, the goal of our paper is to develop an uncertainty measure that encompasses both types of uncertainty for realistic open-world recognition. A good measure should be high in presence of either class confusion or lack of evidence. This is the case for KLoS while current baselines acknowledge only for class confusion, e.g. entropy, (see also Fig. 1) or lack of evidence, e.g. EPKL.
> - Additionally, one can detect the source of uncertainty in KLoS scores by using the decomposition in Eq. (6). In Appendix B, we show that the negative log-likelihood term relates to class confusion while the remainder helps to detect OOD samples, as illustrated on the synthetic data in Figure 9.
>
> &nbsp;
>
> **W3 - Ablation**
>
> In our experiment, we observed that the initialization of KLoSNet helped to converge faster but also to improve misclassification detection performances. We intuit this is due to two factors: (1) the confidence network is intimately linked to the classifier as it aims to predict a value derived from its output, and (2) the nature of the regression, which may be difficult due to the wide range of values (high for misclassifications, close to zeros for correct predictions).
>
> &nbsp;
>
> **W4 - Minor Concerns**
>
> In the synthetic experiment, we pointed out that Mahalanobis is less suited to detect misclassifications than KLoS as Mahalanobis does not discriminate points close to the decision boundaries from points with a similar distance to the origin (page 6). Hence, the following sentence about “troubles detecting misclassifications” refers to the comparison with KLoS, which is also empirically verified in the image experiments in Table 1. We will fix this sentence in the updated version to avoid confusion.

---

> > ### Comment · Reviewer_FYgq · 2021-11-29
> > **Clarifications**
> >
> > Thank you for the clarifications. I have read through the other reviews and responses and find while there are still some weaknesses, most of the concerns are sufficiently adressed. I have raised my rating.

---

### Author Response · Authors · 2021-11-17
**General response and update**

Dear reviewers, we would like to thank you for your careful reading of our work and your detailed comments.. Incorporating the reviewer’s feedback, we have uploaded a revised manuscript including new results with relative Mahalanobis baseline in Appendix D.5 and refactoring some sentence to avoid possible confusions.

We hope our responses and revisions address all reviewers’ concerns, and we would greatly appreciate any further comments and clarifications that we can make.

---

### Decision · Program_Chairs · 2022-01-20

**Decision:**

Reject

**Comment:**

This submission received 4 diverging ratings: 6, 5, 5, 3. On the positive side, reviewers appreciated the central idea and a quality manuscript. At the same time, they have raised important concerns around unfair comparisons with baselines, experiments not fully supporting the claims and lack of comparisons with some prior methods. After discussions with the authors most reviewers stayed with their original ratings.
The AC agrees that the weaknesses in this case outweigh the strengths. The final recommendation is to reject.